# KINDLE: Knowledge-Guided Distillation for Prior-Free Gene Regulatory Network Inference

**Rui Peng[1,2], Yuchen Lu[3], Qichen Sun[1], Yuxing Lu[1], Chi Zhang[1], Ziru Liu[4], Jinzhuo Wang[1]***

[1]Department of Big Data and Biomedical AI, College of Future Technology, Peking University
[2]Center for BioMed-X Research, Academy for Advanced Interdisciplinary Studies, Peking University
[3]School of Physics, Peking University
[4]Yuanpei College, Peking University
{pengrui,luyuchen2002,2000010820,luyx,cszc21,lzr}@stu.pku.edu.cn
wangjinzhuo@pku.edu.cn

## Abstract

Gene regulatory network (GRN) inference serves as a cornerstone for deciphering cellular decision-making processes. Early approaches rely exclusively on gene expression data, thus their predictive power remain fundamentally constrained by the vast combinatorial space of potential gene-gene interactions. Subsequent methods integrate prior knowledge to mitigate this challenge by restricting the solution space to biologically plausible interactions. However, we argue that the effectiveness of these approaches is contingent upon the precision of prior information and the reduction in the search space will circumscribe the models' potential for novel biological discoveries. To address these limitations, we introduce KINDLE, a three-stage framework that decouples GRN inference from prior knowledge dependencies. KINDLE trains a teacher model that integrates prior knowledge with temporal gene expression dynamics and subsequently distills this encoded knowledge to a student model, enabling accurate GRN inference solely from expression data without access to any prior. KINDLE achieves state-of-the-art performance across four benchmark datasets. Notably, it successfully identifies key transcription factors governing mouse embryonic development and precisely characterizes their functional roles. In mouse hematopoietic stem cell data, KINDLE accurately predicts fate transition outcomes following knockout of two critical regulators (Gata1 and Spi1). These biological validations demonstrate our framework's dual capability in maintaining topological inference precision while preserving discovery potential for novel biological mechanisms.

## 1 Introduction

Gene regulatory network (GRN) represents a directed graph that depicts the regulatory interactions between genes, where nodes consist of transcription factors (TFs) and target genes (TGs). A directed edge between a TF and a TG signifies the TF's capacity to bind the cis-regulatory elements of the TG and subsequently modulates its transcriptional activity [1]. GRN provides mechanistic blueprints for understanding regulatory logic underlying lineage commitment, maintenance, and reprogramming [2]. Precisely resolved GRN enables mechanistic interpretations of lineage bifurcation, aging process, and tumor-related dysregulation [3].

Despite their biological significance, GRN inference remains technically challenging. Early inference methods that rely solely on gene expression data face inherent limitations: The explorable TF-TG

---

*Corresponding Author

39th Conference on Neural Information Processing Systems (NeurIPS 2025).

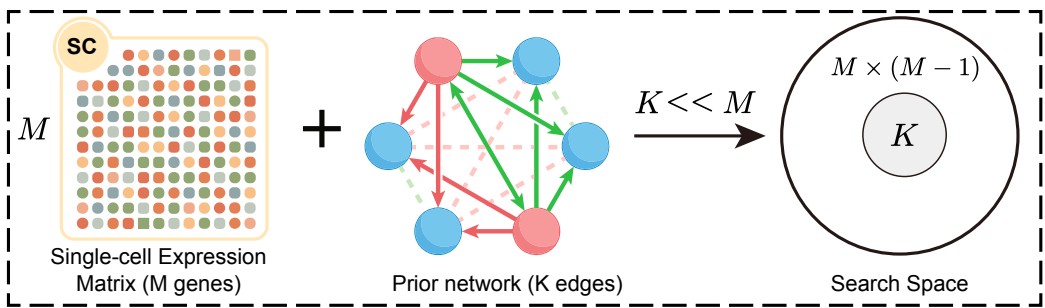

Figure 1: For a dataset with $M$ genes, the search space for gene pairs spans $M \times (M-1)$ possible interactions, as self-loop edges are not considered. Prior-based methods drastically narrow the exploration to the $K$ edges supported by prior knowledge ($K \ll M$).

interaction space scales quadratically with the number of genes, resulting in approximately 1 billion potential regulatory interactions within the whole genome (comprising approximate 30,000 genes). This vast search space fundamentally constrains the performance of expression-based approaches. Contemporary methods address this by incorporating prior knowledge from complementary data (e.g., scATAC-seq [4] or Hi-C [5]) to constrain the search space to pre-defined TF-TG pairs, as illustraed in Figure 1. Although prior-based approaches enhance performance by narrowing the search space, they impose two major limitations. Firstly, with a fixed prior network, an algorithm is confined to searching among its existing edges, and its performance depends on the overlap between the prior and the ground truth network. A perfect match allows for 100% accuracy, while minimal or no overlap leads to zero accuracy for all algorithms. Secondly, limiting candidate edges to the prior prevents the detection of regulatory interactions absent from it, a critical drawback that fundamentally constrains a model's utility for scientific discovery. For example, analyzing gene expression data from cancer cells might reveal a previously unknown transcription factor regulating a pro-oncogenic gene network. Such a discovery, which is impossible when confined to a prior network of already-validated interactions, could lead to the development of new precision therapies.

To overcome prior-dependent limitations, we propose a strategy inspired by learning with privileged information [6]. This paradigm obtains a teacher model using supplementary privileged features during training, followed by knowledge transfer to a student model operating without access to such features. Building on this framework, we develop a three-stage architecture named KINDLE (**K**nowledge-gu**I**ded **N**etwork **D**isti**L**lation for prior-free GRN inf**E**rence) to infer accurate GRN without relying on prior information. The first stage trains a teacher model integrating both gene expression data and external priors. Notably, inspired by TRIGON's temporal causality modeling [7], the teacher model explicitly captures temporal regulatory dynamics by predicting future gene expression states from historical expression profiles, rather than relying on static gene co-expression analysis. The incorporated prior knowledge further refines the candidate regulatory space, generating temporally coherent and biologically plausible regulatory maps. The second stage implements knowledge distillation to train a student model through teacher supervision while completely eschewing prior information. The final stage deploys the student model for prior-independent GRN inference using expression data exclusively, thereby achieving scalable and unbiased reconstruction of regulatory networks. Our contributions are summarized as follows:

- We propose KINDLE to eliminate prior dependence in GRN inference by knowledge distillation, which achieves state-of-the-art performance across four benchmark datasets without requiring prior knowledge.

- On mouse embryonic stem cell development data, KINDLE successfully identifies key TFs and predicts their functional roles during differentiation processes.

- For mouse hematopoietic stem cell development, KINDLE accurately predicts the effects of Gata1 and Spi1 knockouts on cell fate determination, demonstrating its capability to capture critical regulatory mechanisms.

## 2 Related work

**Prior-based GRN inference.** Early attempts in GRN inference predominantly relied on co-expression analyses from bulk or single-cell transcriptomic datasets [8–13]. However, the inherent limitations of unimodal data approaches became evident due to the vast combinatorial search space of potential TF–TG interactions, which severely restricted their predictive performance. To constrain the solution space, contemporary computational pipelines strategically integrated external biological priors during model optimization. For instance, LINGER [14] employed a neural network architecture trained on paired single-cell RNA-seq and ATAC-seq profiles to predict gene expression dynamics through systematic integration of TF abundance and chromatin accessibility. CEFCON [15] implemented a graph attention network initialized with motif-informed adjacency matrix, synergistically coupling cell lineage-specific GRN inference with network control theory. The Celloracle framework [16] operationalized promoter-enhancer interaction maps coupled with DNA motif annotations to establish a base GRN architecture, which undergoes iterative refinement through ridge regression. While demonstrating methodological innovation, all these approaches exhibited fundamental dependence on the precision and comprehensiveness of incorporated prior knowledge, inaccuracies in prior specification risk propagating systematic biases.

**Privileged-feature distillation.** Privileged-feature distillation uses auxiliary data accessible exclusively during training while eliminating their requirement during downstream deployment. In this paradigm, a teacher model with privileged input creates informative soft targets or latent representations to supervise a student model restricted to privileged input. Theoretical analyses in learning-to-rank contexts showed that balancing data-driven loss and teacher guidance helps distilled students outperform non-distilled models [6]. Empirically, the BLEND computational framework [17] validated this by applying the methodology to large-scale neurobiological datasets, where behavioral trajectory data acted as privileged supervisory signals during teacher model optimization, and the distilled neural-activity-only student exceled in population coding decryption tasks. Overall, these theoretical and applied advancements established privileged-feature distillation as a robust way to eliminate the dependence on noisy or resource-intensive prior information, a critical but underexploited property with potential to enhance GRN inference.

## 3 Methodology

### 3.1 Theoretical Foundation

Our framework is grounded on the causal hypothesis that GRN intrinsically govern transcriptional dynamics through time-evolving interactions. Formally, let $\mathbf{G} \in \mathbb{R}^{N \times M}$ denotes the temporal single-cell expression matrix, where $N$ represents temporally ordered cellular states and $M$ is the number of genes. We posit that an accurate GRN adjacency matrix $\mathbf{A} \in \mathbb{R}^{M \times M}$ should encode sufficient mechanistic information to predict future expression states from historical observations and thus satisfy:

$$\mathbf{G}_{T+1:T+W} \approx \mathcal{F}(\mathbf{G}_{1:T}, \mathbf{A}) \tag{1}$$

where $\mathcal{F}$ encodes the nonlinear regulatory kinetics, $T$ and $W$ define historical and future time windows respectively. The GRN inference problem is thus reframed as learning a minimal sufficient interaction matrix $\mathbf{A}^*$ that minimizes the difference between predicted and actual gene expression profiles:

$$\mathbf{A}^* = \arg\min_{\mathbf{A}} \|\mathcal{F}(\mathbf{G}_{1:T}, \mathbf{A}) - \mathbf{G}_{T+1:T+W}\|_2^2 \tag{2}$$

We present KINDLE to operationalize this theory and infer the prior-free GRN by three sequential phases: initial supervised training of the teacher model to assimilate prior network guidance, subsequent distillation of the teacher's regulatory insight into a lightweight student model, and ultimately deployment of the prior independent student model for high-fidelity GRN inference.

### 3.2 Teacher Model

As illustrated in Figure 2, the teacher model is equipped with hierarchical attention mechanisms, consisting of temporal and spatial layers. In the temporal layer, a lower triangular mask is applied to the attention weights, ensuring each gene's expression at time step $t$ only attends to its historical

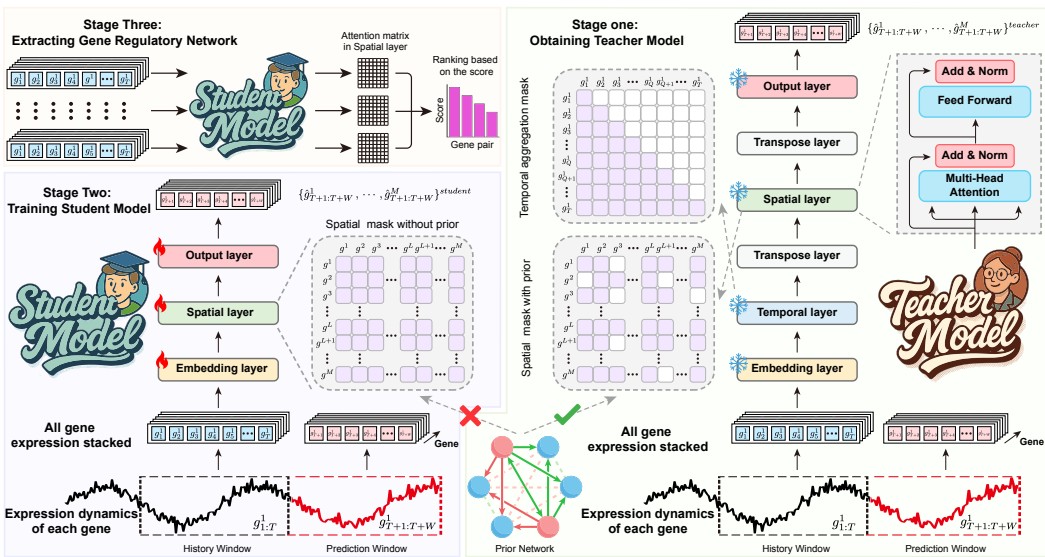

Figure 2: Illustration of KINDLE framework. The pipeline consists of three stages: (1) Teacher model integrates prior knowledge to learn causal regulatory relationships via explicit modeling of expression state transitions across time windows. (2) With teacher model parameters frozen, knowledge distillation transfers regulatory insights to a lightweight student model that operates exclusively on expression data, free of prior inputs. (3) The trained student model is deployed to infer GRN, yielding prior-decoupled network that maintain high accuracy.

states $\{1, ..., t-1\}$. This causal constraint mirrors the irreversibility of cellular differentiation, where progenitor cells cannot access transcriptional information from their descendants. The spatial layer employs a prior-derived binary mask $\mathbf{M}^{spatial} \in \{0, 1\}^{M \times M}$, where $\mathbf{M}_{ij}^{spatial} = 1$ indicates a documented regulatory interaction from gene $i$ to $j$ in prior knowledge. This mask sparsifies attention computation by restricting cross-gene interactions to curated regulatory pairs, effectively pruning unvalidated relationships while preserving interpretability. Architecturally, the temporal layer processes input tensors $\mathbf{X} \in \mathbb{R}^{B \times T \times M}$ (batch size $B$, time steps $T$, genes $M$) and outputs a tensor of identical dimensions. To enable gene-centric regulatory modeling in the subsequent spatial layer, we perform axis transposition $\mathbb{R}^{B \times T \times M} \rightarrow \mathbb{R}^{B \times M \times T}$, restructuring the tensor to treat each gene's temporal trajectory as an independent sequence. This dimensional reorganization permits parallelized computation of gene-specific attention weights across all $M$ genes while maintaining temporal dependencies. Following spatial attention computation, the tensor undergoes transposition operation $\mathbb{R}^{B \times M \times T} \rightarrow \mathbb{R}^{B \times T \times M}$, followed by linear projection to $\mathbb{R}^{B \times W \times M}$ for $W$-step gene expression prediction. The end-to-end framework optimizes regulatory dynamics by minimizing the mean squared error:

$$\mathcal{L}_{\text{teacher}} = \frac{1}{B \cdot W \cdot M} \sum_{b=1}^{B} \sum_{w=1}^{W} \sum_{m=1}^{M} \left\| \hat{\mathbf{Y}}_{t+w,m}^{(b)} - \mathbf{Y}_{t+w,m}^{(b)} \right\|_2^2 \tag{3}$$

where $\hat{\mathbf{Y}}$ and $\mathbf{Y}$ denote predicted and ground truth expression matrix respectively, indexed by batch $b$, forecast window $w$, and gene $m$. While the attention matrix $\mathbf{A} \in \mathbb{R}^{M \times M}$ extracted from teacher model's spatial layer during inference could be a prior-constrained approximation of the theoretically optimal matrix $\mathbf{A}^*$ defined in Eq.2, the solution remains fundamentally constrained by its prior-dependent architecture. Specifically, the binary masking operation irreversibly eliminates attention weights for gene pairs absent in the prior knowledge (i.e., positions where $\mathbf{M}_{ij}^{spatial} = 0$), thereby restricting the teacher model's attention exclusively to a sparse subset of regulatory interactions defined by prior-informed positions (i.e., $\mathbf{M}_{ij}^{spatial} = 1$). This prior-induced myopia severely limits applicability to emerging biological systems with incomplete interactome annotations. To overcome this fundamental limitation, we design a student model that learns the teacher's regulatory knowledge through distillation without inheriting its prior constraints.

### 3.3 Student Model

We formalize the student model as $f_{\theta_S} \in \{f | f : \mathbf{G}_{1:T} \mapsto \mathbf{G}_{T+1:T+W}\}$, operating exclusively on raw expression matrix $\mathbf{G}_{1:T} \in \mathbb{R}^{B \times T \times M}$ without prior network integration. We let $f_{\theta_T}$ be the teacher model and the parameter optimization of the student model aims to minimize the following loss:

$$
\alpha \cdot \underbrace{\sum_{b,w,m} \|f_{\theta_S}(\mathbf{G}_{1:T})_{b,m} - \mathbf{G}_{b,T+w,m}\|_2^2}_{\text{Prediction Loss}} + (1-\alpha) \cdot \underbrace{\sum_{b,m} \mathcal{L}_{\text{distill}}\left(f_{\theta_S}(\mathbf{G}_{1:T})_{b,m}, f_{\theta_T}(\mathbf{G}_{1:T}, \mathbf{M}^{spatial})_{b,m}\right)}_{\text{Regulatory Distillation Loss}}
$$
(4)

The hyperparameter $\alpha \in (0,1)$ governs the trade-off between expression prediction fidelity and regulatory knowledge transfer. Crucially, as diagrammed in Figure 2, the student architecture implements two critical modifications: (1) Elimination of the prior-dependent spatial mask $\mathbf{M}^{\text{spatial}}$ in attention computation, enabling unrestricted interaction modeling between all gene pairs. (2) Removal of the teacher's temporal layer while preserving temporal causality through distillation, resulting in a lightweight model.

Distinct formulations of the distillation loss $\mathcal{L}_{\text{distill}}$ can extract different dimensions of the teacher model's knowledge. In the course of this research, we explore four primary distillation strategies within our KINDLE framework. Each of these strategies is meticulously designed to convey specific aspects of the teacher model's knowledge to the student model, thereby enhancing the latter's performance and understanding.

**Hard Distillation.** We optimize this baseline through a predictive congruence objective, where $\mathcal{L}_{\text{distill}}$ is constructed as direct predictive alignment. Specifically, the framework achieves this by optimizing the squared L2-norm divergence between the teacher's terminal predictions and the student's corresponding outputs, enforcing knowledge transfusion via deterministic supervision of final-layer activations:

$$
\mathcal{L}_{\text{distill}}\left(f_{\theta_S}(\mathbf{G}_{1:T})_{b,m}, f_{\theta_T}(\mathbf{G}_{1:T}, \mathbf{M}^{spatial})_{b,m}\right) = \left\|f_{\theta_S}(\mathbf{G}_{1:T})_{b,m} - f_{\theta_T}(\mathbf{G}_{1:T}, \mathbf{M}^{spatial})_{b,m}\right\|_2^2
$$
(5)

**Soft Distillation.** This paradigm implements probabilistic knowledge transfer through entropy-regulated distribution matching. The framework introduces temperature parameter $\tau$ to soften the logits before applying the softmax function, formally expressed as:

$$
\mathcal{L}_{\text{distill}}\left(f_{\theta_S}(\mathbf{G}_{1:T})_{b,m}, f_{\theta_T}(\mathbf{G}_{1:T}, \mathbf{M}^{spatial})_{b,m}\right) = \text{KL}\left(\sigma\left(\frac{f_{\theta_S}(\mathbf{G}_{1:T})_{b,m}}{\tau}\right) \middle\| \sigma\left(\frac{f_{\theta_T}(\mathbf{G}_{1:T}, \mathbf{M}^{spatial})_{b,m}}{\tau}\right)\right)
$$
(6)

where $\sigma$ is the softmax function, and $\text{KL}(\cdot\|\cdot)$ is the Kullback-Leibler divergence.

In addition to the aforementioned hard target distillation and soft probabilistic matching, we develop correlation distillation to preserve structural dependencies in feature representations. The core objective is to align the teacher-student correlation manifolds through kernel-induced similarity measures, formalized as:

$$
\mathcal{L}_{\text{distill}}\left(f_{\theta_S}(\mathbf{G}_{1:T})_{b,m}, f_{\theta_T}(\mathbf{G}_{1:T}, \mathbf{M}^{spatial})_{b,m}\right) = \mathcal{K}(f_{\theta_S}(\mathbf{G}_{1:T})_{b,m}, f_{\theta_T}(\mathbf{G}_{1:T}, \mathbf{M}^{spatial})_{b,m})
$$
(7)

where $\mathcal{K}(\cdot, \cdot)$ denote kernel methods to compute the correlation between output of $f_{\theta_S}$ and $f_{\theta_T}$. To address the challenges posed by the high dimensionality of embedded feature spaces in analyzing complex inter-instance correlations, we propose two different kernel methods to effectively capture the high-order correlations between instances within the feature space.

**Bilinear Pool.** It computes inter-instance correlations through outer product operations. Formally, the Bilinear Pool kernel is defined as:

$$
\mathcal{K}_{\text{bilinear}}(f_{\theta_S}(\mathbf{G}_{1:T})_{b,m}, f_{\theta_T}(\mathbf{G}_{1:T}, \mathbf{M}^{spatial})_{b,m}) = (f_{\theta_S}(\mathbf{G}_{1:T})_{b,m})^\top (f_{\theta_T}(\mathbf{G}_{1:T}, \mathbf{M}^{spatial})_{b,m})
$$
(8)

**Gaussion RBF.** This non-linear operator characterizes instance relationships through exponentially decaying similarity metrics, possessing stronger non-linear manifold learning capabilities compared

Table 1: Comparison of the proposed KINDLE framework with other GRN inference methods on four datasets provided by BEELINE [18]. **Bold values** denote the best performance for the corresponding metric.

| Methods | mESC | | | mHSC-E | | | mHSC-L | | | mHSC-GM | | |
|---|---|---|---|---|---|---|---|---|---|---|---|---|
| | AUROC | AUPRC | F1 | AUROC | AUPRC | F1 | AUROC | AUPRC | F1 | AUROC | AUPRC | F1 |
| **Without Prior** | | | | | | | | | | | | |
| **GRNBoost2 [8]** | 0.537 | 0.127 | 0.203 | 0.397 | 0.034 | 0.087 | 0.515 | 0.181 | 0.297 | 0.474 | 0.083 | 0.146 |
| **GENIE3 [9]** | 0.545 | 0.137 | 0.218 | 0.381 | 0.042 | 0.108 | 0.486 | 0.183 | 0.322 | 0.437 | 0.078 | 0.162 |
| **Random** | 0.506 | 0.083 | 0.152 | 0.493 | 0.087 | 0.161 | 0.518 | 0.135 | 0.227 | 0.504 | 0.083 | 0.154 |
| **Prior Guided** | | | | | | | | | | | | |
| **CEFCON [15]** | 0.479 | 0.253 | 0.429 | 0.531 | 0.405 | 0.551 | **0.653** | 0.659 | 0.675 | 0.457 | 0.444 | 0.647 |
| **Celloracle [16]** | 0.490 | 0.177 | 0.305 | 0.536 | 0.290 | 0.420 | 0.557 | 0.277 | 0.368 | 0.487 | 0.243 | 0.401 |
| **NetREX [19]** | 0.522 | 0.128 | 0.217 | 0.511 | 0.117 | 0.211 | 0.520 | 0.177 | 0.282 | 0.526 | 0.144 | 0.219 |
| **Prior_Random** | 0.498 | 0.318 | 0.482 | 0.492 | 0.389 | 0.570 | 0.522 | 0.551 | 0.691 | 0.509 | 0.464 | 0.627 |
| **KINDLE** | | | | | | | | | | | | |
| **KINDLE (Soft distillation)** | 0.747 | 0.636 | 0.519 | 0.561 | 0.559 | 0.691 | 0.599 | 0.670 | 0.752 | 0.562 | 0.789 | 0.864 |
| **KINDLE (Hard distillation)** | 0.753 | 0.643 | 0.526 | 0.564 | 0.578 | 0.711 | 0.599 | 0.669 | 0.757 | 0.569 | 0.793 | 0.871 |
| **KINDLE (Bilinear Pool)** | 0.751 | 0.644 | 0.521 | 0.551 | 0.574 | 0.723 | 0.567 | 0.581 | 0.761 | 0.561 | 0.787 | 0.867 |
| **KINDLE (Gaussian RBF)** | **0.757** | **0.646** | **0.529** | **0.594** | **0.601** | **0.731** | 0.600 | **0.672** | **0.763** | **0.570** | **0.799** | **0.875** |

to bilinear methods. The kernel admits low-rank Taylor series approximation while preserving topological structures in feature space. Formally, the Gaussion RBF kernel is defined as:

$$\mathcal{K}_{\text{gaussion}}\left(f_{\theta_S}(\mathbf{G}_{1:T})_{b,m}, f_{\theta_T}(\mathbf{G}_{1:T}, \mathbf{M}^{spatial})_{b,m}\right) = exp\left(-\frac{\|f_{\theta_S}(\mathbf{G}_{1:T})_{b,m} - f_{\theta_T}(\mathbf{G}_{1:T}, \mathbf{M}^{spatial})_{b,m}\|_2^2}{2\lambda^2}\right) \quad (9)$$

where $\lambda$ is a hyperparameter that controls the width of the gaussian function.

### 3.3.1 GRN Inference

Given the input gene expression time series $\mathbf{G} \in \mathbb{R}^{N \times M}$, where $N$ denotes the temporal sequence length and $M$ represents the number of genes, we partition the sequence into segments of length $T \in \mathbb{N}^+$. Under the divisibility condition $T \mid N$, we obtain $H = \frac{N}{T}$ non-overlapping samples $\{\mathcal{S}^{(g)}\}_{g=1}^H$, each containing $T$ consecutive temporal observations:

$$\mathcal{S}^{(g)} = \mathbf{G}_{(T \cdot (g-1)+1):(T \cdot g)} \quad \forall g \in \{1, ..., H\} \quad (10)$$

For each partitioned sample $\mathcal{S}^{(g)}$, the student model $f_{\theta_S}$ generates attention matrix $\mathbf{A}^{(g)} \in \mathbb{R}^{M \times M}$ in its spatial layer. We compute the optimal approximation $\hat{\mathbf{A}}$ to the theoretical $\mathbf{A}^*$ in Eq.2 through temporal ensemble:

$$\hat{\mathbf{A}} = \frac{1}{H} \sum_{g=1}^H \mathbf{A}^{(g)} \quad (11)$$

The final GRN $\mathcal{G}_{pred}$ is established by ranking the edge weights in the matrix $\hat{\mathbf{A}}$ and selecting the top-$k$ most significant connections, where $k$ corresponds exactly to the number of edges in the ground truth regulatory network $\mathcal{G}_{gt}$ provided with each benchmarking dataset:

$$\mathcal{G}_{pred} = \{(i,j) \mid \hat{A}_{ij} \in \text{Top}_k(\hat{A})\}, \quad k = |\mathcal{G}_{gt}| \quad (12)$$

The detailed pseudocode implementations of KINDLE are provided in Appendix F.

## 4 Experiments

### 4.1 KINDLE Achieved State-of-the-Art Performance in GRN Benchmarks

The evaluation of KINDLE strictly adheres to the benchmarking protocol introduced in BEELINE [18]. We systematically validated our approach on four mouse differentiation datasets, the embryonic stem cell (mESC) dataset as well as three hematopoietic lineages: Erythrocyte (mHSC-E), Granulocyte-Monocyte (mHSC-GM), and Lymphocyte (mHSC-L). Following BEELINE's established framework, we treated GRN inference as a binary classification task, employing lineage-matched reference GRN derived from ChIP-seq experiments as ground truth (see Appendix B.1 for detailed information). Performance was quantified through three metrics: area under the receiver operating characteristic curve (AUROC), area under the precision-recall curve (AUPRC), and F1 score (see Appendix B.2 for pseudocode of metric calculation). KINDLE was compared against seven competitive baselines: expression-based methods (GENIE3 [9], GRNBoost2 [8]), prior-based

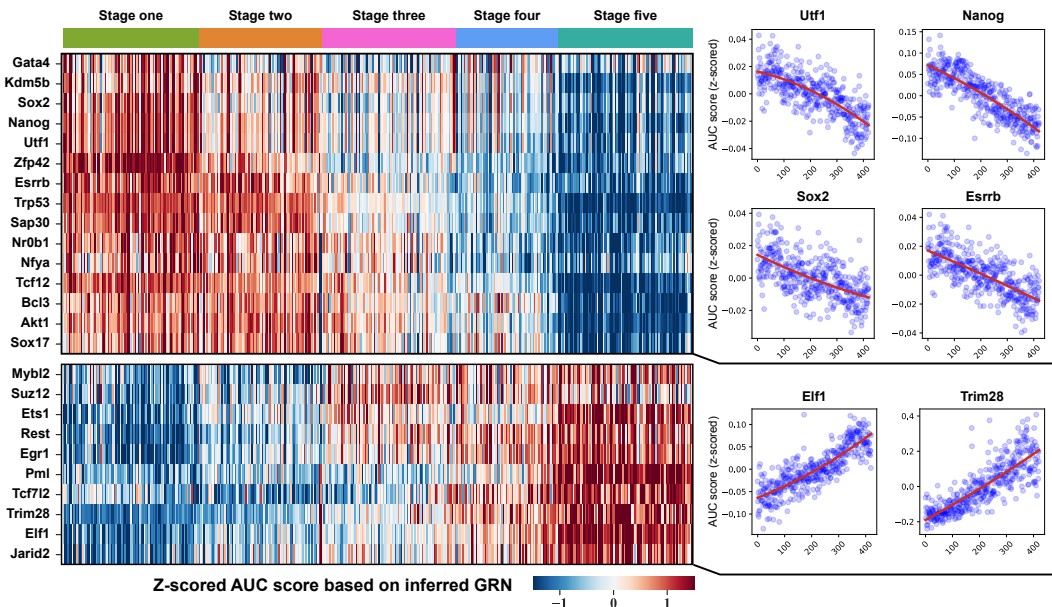

Figure 3: Temporal dynamics of TF regulatory performance during mouse embryonic stem cell differentiation. **Left:** Heatmap visualization of z-scored AUC scores reveals bimodal temporal patterns through hierarchical clustering. Two distinct TF clusters emerge, demonstrating their divergent regulatory roles during differentiation. **Right:** Trend analysis of AUC score, with developmental time points on x-axis and z-scored AUC score on y-axis. Red curves represent quadratic polynomial fits to the dynamic profiles. Complete regression curves for all 25 TFs are available in Appendix E.

approaches (CEFCON [15], CellOracle [16], NetREX [19]), and two random controls (Random, Prior_Random). Detailed descriptions of datasets, ground truth networks, and baseline models are provided in Appendix A.1, Appendix A.2, and Appendix A.3 respectively.

As shown in Table 1, all four KINDLE variants demonstrated substantial improvements over baselines despite requiring no external biological priors. The soft distillation variant, representing our weakest configuration, surpassed the best expression-based method (GENIE3) by 0.499, 0.517, 0.490, 0.711 in AUPRC across datasets and outperformed CellOracle by 0.546 in mHSC-GM. Among variants, the one with Gaussian RBF (hereafter **KINDLE-Gaussian**) achieved the best performance in 11 of 12 dataset-metric combinations. On the mESC dataset, KINDLE-Gaussian improved AUROC from 0.545 (GENIE3) to 0.757 (39% increase), elevated AUPRC from 0.253 (CEFCON) to 0.646 (155% improvement), and raised F1 score from 0.429 to 0.529 (23% gain). Comparable enhancements emerged in hematopoietic lineages: AUPRC increased by 48% (0.405 → 0.601) for erythroid differentiation and nearly doubled (0.444 → 0.799) in granulocyte-monocyte development, accompanied by a 0.228 absolute F1 score improvement.

Notably, KINDLE's superiority proved most pronounced in AUPRC and F1 metrics. As summarized in Table 2, the validated edges in ground truth network constituting merely 0.9% (mESC), 0.65% (mHSC-E), 0.7% (mHSC-GM), and 1.15% (mHSC-L) of candidate edges, thus introducing severe class imbalance in the binary classification task. Under such conditions, AUPRC and F1 serve as more reliable performance indicators than AUROC (detailed justifications are provided in Appendix B.3). Therefore, the consistent AUPRC and F1 improvements demonstrated that privileged knowledge distillation provided a robust, prior-free route to GRN inference, outperforming not only expression-only algorithms but also methods that rely on explicit biological priors.

### 4.2 KINDLE Identified Key TFs and Their Stage-Specific Functions

Beyond quantitative metrics for assessing GRN accuracy, a crucial evaluation criterion lies in determining whether the inferred GRN can identify key TFs that orchestrate differentiation processes. SCENIC [20] introduced the AUCell algorithm, which calculates AUC scores for TF regulon (the set of all target genes of a TF in the inferred GRN) through rank-based enrichment analysis. This score reflects the functional activity of the TF regulon within each cell, enabling systematic identification

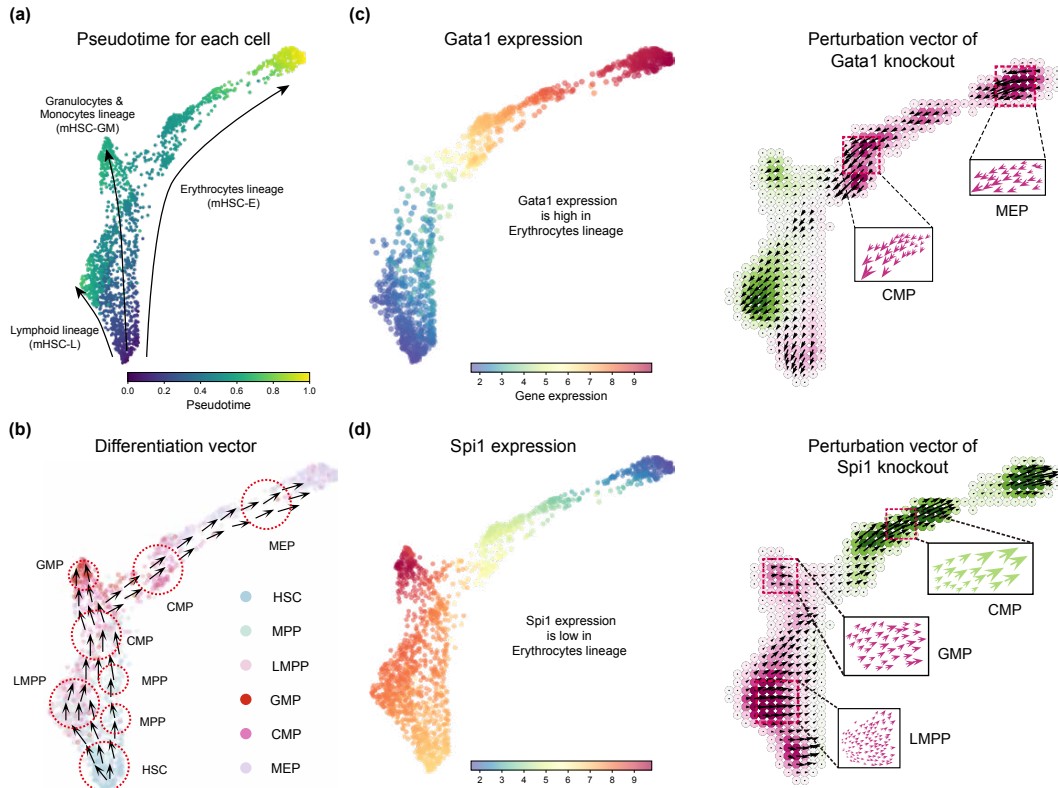

Figure 4: In silico perturbation analysis validates that KINDLE can accurately predict the cell fate transitions during the multi-lineage haematopoietic differentiation. (A) Slingshot-derived pseudotime trajectory embedding. (B) Cell-type annotations overlaid on the pseudotime landscape. (C) **Left:** Expression profiles of Gata1. **Right:** Silencing Gata1 induces reverse perturbation vectors along the erythroid branch and leads to differentiation arrest in CMP and MEP. (D) **Left:** Expression profiles of Spi1. **Right:** Silencing Spi1 represses the differention of LMPP and GMP and promotes erythroid progression in CMP.

of key regulators that drive cellular state transitions (see Appendix C for detailed methodology of AUCell). Following this protocol, we computed AUC scores from the GRN inferred by KINDLE-Gaussian to estimate per-cell TF activities in the mESC dataset. Given the five differentiation stages in this dataset (see Appendix A.1), we performed analysis of variance on AUC scores to assess whether they varied significantly throughout the differentiation stages and defined key TFs as those with Benjamini–Hochberg-adjusted P-values < 0.01. We identified 25 key TFs, with their adjusted P-values listed in Table 3. Notably, 18 (72%) of 25 identified regulators have established roles in mESC differentiation according to prior literature [21–41].

Temporal patterning of the 25 TF regulon activities was investigated through hierarchical clustering of z-scored AUC scores (Figure 3). Two anti-correlated activation modules emerged from this unsupervised analysis: **Early-stage regulators** (Nanog, Sox2, Nr0b1, etc.) demonstrated peak activity at stage one with progressive attenuation through subsequent stages. This temporal trend aligns with known biological functions, such as Nanog and Sox2 being highly expressed in the early stage of mouse embryonic stem cells, maintaining the pluripotency of stem cells [42]. Their expression rapidly decreases as cells commit to differentiation, reflecting their pivotal function in regulating the transition from a pluripotent state to more specialized lineages [24]. **Late-stage regulators** (Gata4, Sox17, Kdm5b, etc.) exhibited minimal initial activity but showed significant activation from stage three onward. These results corroborate established mechanisms of lineage specification, where Sox17 overexpression upregulates a set of endoderm-specific gene markers and induces an ESC differentiation program towards primitive endoderm [43]. The emergence of these antiphasic expression patterns demonstrated that KINDLE not only recovered biologically relevant TFs but also assigned each regulator to its stage-specific functional context, thereby elucidating the sequential deployment of transcriptional programs during mESC differentiation.

### 4.3 KINDLE Predicted Lineage-Specific Fate Changes in In-Silico Perturbation

Following the precise identification of key TFs, a practical application involves interrogating their functional roles through systematic perturbation. We employed the Celloracle framework [16] to implement in silico perturbation analysis. This approach simulates TF knockout by setting target TF expression to zero and propagating the perturbation signal through the GRN's topological structure to its target genes, ultimately generating a two-dimensional perturbation vector for each cell that predicts its fate trajectory under the specified perturbation (see Appendix D for algorithmic details). To validate the biological relevance of KINDLE-Gaussian's predictions, we applied this methodology to the mHSC dataset, an ideal and complex benchmark system containing six distinct cell types (HSC, MPP, LMPP, GMP, CMP, MEP; see Appendix A.1 for cell type information) organized along three differentiation trajectories (Figure 4 a,b). The sequential differentiation order of different cell types is shown in Figure 5.

We focused on two well-characterized regulators governing hematopoietic lineage commitment, Gata1 and Spi1. Consistent with their established roles, Gata1 expression dominated in erythroid-lineage cells (CMP and MEP, Figure 4c), while Spi1 showed myeloid-lineage enrichment (LMPP and GMP, Figure 4d). Following the perturbation of Gata1, we generated perturbation vectors for each cell. Notably, the vectors for all cells within the erythroid lineage were opposite to the developmental direction of pseudotime trajectory shown in Figure 4a. This observation indicated that in the absence of Gata1, cells tend to revert to earlier progenitor states rather than progress towards more mature cell identities. To quantitatively assess the perturbation effect, we calculated a perturbation score for each cell and coloured the cells (purple $\rightarrow$ negative score, differentiation inhibited; green $\rightarrow$ positive score, differentiation promoted, see Appendix D.3 for additional details of perturbation score calculation). In erythroid lineage, all cells received negative scores, with the strongest inhibitory effect concentrated in CMP and MEP, the cell populations with the highest Gata1 expression levels. Subsequently, we applied the same perturbation procedure to Spi1. Upon silencing Spi1, all cells showed a developmental trajectory towards the erythroid lineage, with CMP differentiation being promoted as well as GMP and LMPP differentiation being inhibited (Figure 4d). These perturbation results are consistent with previous reports [44–47], where Gata1 promotes the differentiation of CMP into MEP (resulting in inhibition of CMP and MEP differentiation when Gata1 knockout), and Spi1 suppresses the CMP to MEP transition (resulting in CMP perturbation vectors pointing towards MEP upon Spi1 silencing).

Collectively, the in silico perturbation analyses demonstrated that within the haematopoietic system, KINDLE accurately modeled the downstream effects of key TFs knockout. Hence, beyond pinpointing key TFs, KINDLE provided a mechanistic scaffold for the rational design of cell-fate-engineering strategies.

### 4.4 Implementation Details

We trained KINDLE on an 80GB Nvidia A100 GPU for 30 epochs with a batch size of 32. To prevent overfitting, we implemented an early stopping strategy with a patience value set to 3. For optimization, we employed the Adam optimizer in conjunction with a warmup strategy, gradually increasing the learning rate from 0 to 1e-4. Subsequently, a CosineAnnealingLR scheduler was utilized to further fine-tune the learning rate. During the training process, we conducted experiments with five distinct values for the hyperparameter $W$ (1, 2, 4, 8, and 16), ultimately selecting $W = 16$ as it yielded the optimal results reported in this paper.

## 5 Discussion and Limitations

KINDLE advances GRN inference methodology by decoupling algorithm from prior knowledge dependency (the longstanding bottleneck in the field). Through integrating temporal causality modeling with knowledge distillation, our framework successfully transfers regulatory insights learned from privileged prior-augmented data to a prior-free student model, enabling KINDLE to achieve state-of-the-art performance across four benchmark datasets. The model's ability to recover key TFs governing lineage specification validates its capacity to capture biologically interactions and the accurate prediction of knockout effects on hematopoietic cell fate transition underscores its potential for elucidating dynamic regulatory mechanisms in development and disease. The framework's prior-independent nature positions it as a versatile tool for studying poorly characterized

systems, such as non-model organisms or emerging pathological states, where reliable prior networks are often unavailable.

Despite its strengths, KINDLE has several limitations. First, its reliance on temporal gene expression data restricts applicability to datasets with longitudinal sampling. Second, the distillation process may inherit biases from the teacher model's prior-dependent training phase, potentially propagating errors from incomplete or noisy priors. Third, the current implementation focuses on transcriptional regulation, omitting post-transcriptional and epigenetic layers of gene regulation that could refine network predictions. Addressing these challenges will be critical for extending the framework's utility across diverse biological contexts.

## Acknowledgments and Disclosure of Funding

This research was supported by National Key Research and Development Program of China (2024YFF0507400) and National Natural Science Foundation of China (6220071694).

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

# A    Supplementary Contents of Datasets and Baselines

## A.1    Datasets

**Mouse embryonic stem cell (mESC).**    The mESC dataset contains Single-cell RNA sequencing (scRNA-seq) expression measurements for 421 primitive endoderm cells differentiated from mESCs, collected at five time points (0, 12, 24, 48, and 72 hours). Pseudotime computation was performed using Slingshot [48], with cells at 0 hours as the starting cluster and cells at 72 hours as the terminal differentiation state.

**Mouse hematopoietic stem cell (mHSC).** The mHSC dataset comprises 1,656 hematopoietic stem and progenitor cells traversing six critical differentiation states: hematopoietic stem cells (HSCs), multipotent progenitors (MPPs), lymphoid-primed multipotent progenitors (LMPPs), common myeloid progenitors (CMPs), megakaryocyte-erythrocyte progenitors (MEPs), and granulocyte-monocyte progeni-

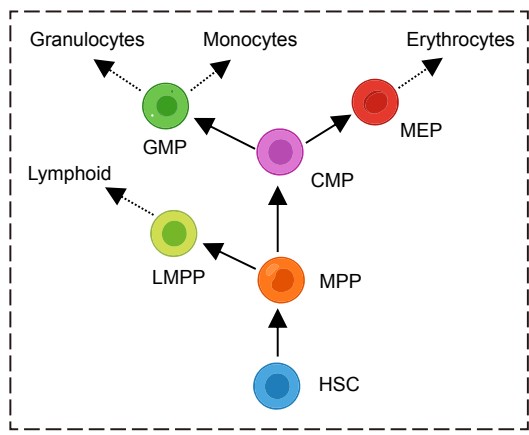

Figure 5: Differentiation schematic of six cell states in the mHSC dataset.

tors (GMPs). As visualized in Figure 5, these cell types follow distinct differentiation trajectories across three developmental lineages. Pseudotime trajectories were computed using the first three principal dimensions derived from DiffusionMap [49]. Gene regulatory networks were independently reconstructed for each lineage.

## A.2    Ground Truth Networks

To benchmark inferred GRN, BEELINE [18] constructed ground truth networks from three stratified categories:

- **Cell-type-specific networks**:
    - Sourced from ENCODE [50], ChIP-Atlas [51], and ESCAPE [52] databases
    - Matched to the scRNA-seq dataset's cell lineage
    - Included *loss-of-function* or *gain-of-function* perturbation data from ESCAPE
- **Non-specific networks**:
    - DoRothEA [53]: Integrated regulatory interactions filtered by confidence levels:
        * Level A: high-confidence ChIP-seq data
        * Level B: likely-confidence interactions
    - RegNetwork [54]: Genome-wide TF–gene and TF–TF interactions across human and mouse
    - TRRUST [55]: Manually curated TF–TG pairs from literature mining
- **Functional networks**:
    - Derived from STRING [56] protein interaction databases
    - Captured indirect regulatory effects (e.g., phosphorylation, co-expression)

Notably, in our study, cell-type-specific networks were employed as the ground truth for the benchmark evaluation of our model.

## A.3    Baselines

**GENIE3 [9].**    GENIE3 decomposes GRN inference into $p$ regression problems for $p$ genes, using tree-based ensembles to quantify regulatory potential. For each target gene, it evaluates the predictive importance of all other genes' expression patterns as putative regulators. These pairwise importance scores are aggregated to rank regulatory interactions and reconstruct directed networks.

**GRNBoost2 [8].** GRNBoost2 is a gradient-boosting-based algorithm for GRN inference. Inspired by GENIE3, it trains tree-based regression models to predict each gene's expression profile using TF expression data. The method employs regularized stochastic gradient boosting with an early-stopping heuristic: training terminates when out-of-bag data indicates non-improving loss function (average improvement < 0). Regulatory associations are aggregated and ranked by importance scores to construct the final GRN.

**CEFCON [15].** CEFCON is a network control theory framework for cell fate analysis using scRNA-seq data. It constructs lineage-specific GRN via graph attention neural network under contrastive learning, aggregating gene interactions through adaptive neighborhood weighting. By integrating minimum feedback vertex sets and minimum dominating sets with GRN influence scores, it identifies driver regulators of cell fate transitions.

**CellOracle [16].** CellOracle integrates scATAC-seq motif analysis and scRNA-seq data to model context-dependent GRNs. It first builds a base network through TF-binding motif scanning of regulatory DNA, and then refines edge weights using regularized linear models on expression data. Through in silico TF perturbation simulations, it predicts cell identity shifts by propagating signals across the GRN and analyzing pseudotime gradient vector fields.

**NetREX [19].** NetREX reconstructs context-specific GRN by optimizing prior networks against expression data. Formulated as a non-convex $l_0$-norm optimization problem, it iteratively modifies network topology using proximal alternative linearized maximization.

**Random.** The Random baseline generates GRN by randomly selecting $k$ edges from all possible gene-gene interactions, where $k$ equals the number of edges in the ground truth GRN.

**Prior_Random.** Prior_Random selects $k$ edges exclusively from prior network interactions ($k$ matches the number of ground truth GRN edges).

# B Supplementary Contents of Benchmark Testing

## B.1 The evaluation of GRN is regarded as a binary classification problem

The evaluation of inferred GRN can be formalized as a binary classification task, where edges in the inferred network are categorized relative to a ground truth GRN. As depicted in the Figure 6, let $\mathcal{G}_{\text{true}} = (V, E_{\text{true}})$ denote the ground truth network, and $\mathcal{G}_{\text{pred}} = (V, E_{\text{pred}})$ represent the inferred network. Each edge $e \in E_{\text{pred}}$ is classified into one of four categories: (1) True Positive (TP) : $e \in E_{\text{pred}} \cap E_{\text{true}}$. (2) False Positive (FP) : $e \in E_{\text{pred}} - E_{\text{true}}$. (3) True Negative (TN) : $e \notin E_{\text{pred}} \cup E_{\text{true}}$. (4) False Negative (FN) : $e \in E_{\text{true}} - E_{\text{pred}}$. Performance metrics are derived as follows: $\text{Precision} = \frac{\text{TP}}{\text{TP}+\text{FP}}$, $\text{Recall} = \frac{\text{TP}}{\text{TP}+\text{FN}}$, $\text{F1} = 2 \cdot \frac{\text{Precision} \cdot \text{Recall}}{\text{Precision} + \text{Recall}}$, $\text{TPR} = \frac{\text{TP}}{\text{TP}+\text{FN}}$, $\text{FPR} = \frac{\text{FP}}{\text{FP}+\text{TN}}$, $\text{AUROC} = \int_0^1 \text{TPR} \, d\text{FPR}$, $\text{AUPRC} = \int_0^1 \text{Precision} \, d\text{Recall}$. These metrics enable systematic comparison of GRN inference methods, quantifying both accuracy and robustness.

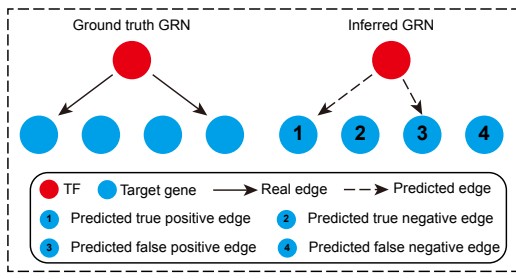

Figure 6: Schematic diagram of GRN evaluation, where the predicted edges can be classified into four types.

## B.2 GRN Benchmakring Algorithm

---

**Algorithm 1** GRN Benchmark Evaluation

---

1: **Input**:
2:    - Ground truth GRN: $\mathcal{G}_{\text{true}} = (V, E_{\text{true}})$
3:    - Predicted GRN for $N$ algorithms: $\{\mathcal{G}_{\text{pred}}^{(i)} = (V, E_{\text{pred}}^{(i)})\}_{i=1}^{N}$
4: **Output**: Metrics $\in \mathbb{R}^{N \times 3}$        ▷ DataFrame containing AUROC, AUPRC, F1
5: // Preprocess ground truth
6: $E_{\text{true}} \leftarrow E_{\text{true}} - \{(v, v) \mid v \in V\}$        ▷ Remove self-loops
7: $E_{\text{true}} \leftarrow \text{Deduplicate}(E_{\text{true}})$        ▷ Remove duplicate edges
8: results $\leftarrow \emptyset$        ▷ Initialize metric collection
9: **for** each predicted GRN $\mathcal{G}_{\text{pred}}^{(i)}$ **do**
10:     $E_{\text{pred}}^{(i)} \leftarrow \text{Sort}(\text{Deduplicate}(E_{\text{pred}}^{(i)}))$        ▷ Sort predicted edges
11:     // Generate candidate edges
12:     $P_{\text{all}} \leftarrow V \times V - \{(v, v)\}$        ▷ All potential non-self edges
13:     $\mathbf{y}_{\text{true}} \leftarrow [\mathbb{I}(e \in E_{\text{true}}) \mid \forall e \in P_{\text{all}}]$    ▷ Obtain true labels, where $\mathbb{I}$ is the indicator function
14:     // Obtain predicted edges
15:     $\mathbf{y}_{\text{pred}}^{(i)} \leftarrow \text{TopK}(E_{pred}^{(i)})$        ▷ Select the top k edges
16:     // Calculate metrics
17:     $\text{TPR}^{(i)}, \text{FPR}^{(i)} \leftarrow \frac{\text{TP}}{\text{TP+FN}}, \frac{\text{FP}}{\text{FP+TN}}$        ▷ ROC components
18:     $\text{Precision}^{(i)}, \text{Recall}^{(i)} \leftarrow \frac{\text{TP}}{\text{TP+FP}}, \frac{\text{TP}}{\text{TP+FN}}$        ▷ PR components
19:     $\text{F1}^{(i)} \leftarrow 2 \cdot \frac{\text{Precision}^{(i)} \cdot \text{Recall}^{(i)}}{\text{Precision}^{(i)} + \text{Recall}^{(i)}}$        ▷ Calculate F1 score
20:     $\text{AUROC}^{(i)}, \text{AUPRC}^{(i)} \leftarrow \int_0^1 \text{TPR} \, d\,\text{FPR}, \int_0^1 \text{Precision} \, d\,\text{Recall}$
21:     results $\leftarrow$ results $\cup \{(i, \text{AUROC}^{(i)}, \text{AUPRC}^{(i)}, \text{F1}^{(i)})\}$        ▷ Record metrics
22: **end for**
23: // Aggregate results
24: Metrics $\leftarrow$ ConstructDataFrame(results)        ▷ Shape: $N \times 3$
25: **return** Metrics

---

## B.3 AUPRC is more robust compared to AUROC

Table 2: The distribution of positive and negative labels in four datasets

|                         | mESC    | mHSC-E  | mHSC-GM | mHSC-L |
|-------------------------|---------|---------|---------|--------|
| Genes                   | 1652    | 1933    | 1520    | 640    |
| Potential edges         | 2727452 | 3734556 | 230880  | 408960 |
| True edges              | 24557   | 24726   | 16198   | 4705   |
| Proportion of true edges| 0.90%   | 0.65%   | 0.70%   | 1.15%  |

The evaluation framework described in Appendix B.2 operates on a search space of TF–TG pairs defined as $\mathcal{E}_{\text{potential}} = M \times (M - 1)$, where $M$ denotes the number of genes in a dataset. Within this space, edges present in the ground truth GRN are defined as $\mathcal{E}_{\text{true}}$. We quantified the distribution of $\mathcal{E}_{\text{potential}}$ and $\mathcal{E}_{\text{true}}$ in Table 2 and found that there is an extreme class imbalance inherent to GRN inference across four datasets. For instance:

$$\text{mESC:} \quad |\mathcal{E}_{\text{potential}}| = 2{,}727{,}452, \quad |\mathcal{E}_{\text{true}}| = 24{,}557 \quad (\sim 0.9\% \text{ positivity rate})$$
$$\text{mHSC-L:} \quad |\mathcal{E}_{\text{potential}}| = 408{,}960, \quad |\mathcal{E}_{\text{true}}| = 4{,}705 \quad (\sim 1.1\% \text{ positivity rate})$$

In such scenarios, AUROC disproportionately emphasizes the majority class (negative edges) due to its reliance on the false positive rate ($\text{FPR} = \frac{\text{FP}}{\text{FP+TN}}$). When $|\mathcal{E}_{\text{true}}| \ll |\mathcal{E}_{\text{potential}}|$, the sum $\text{FP} + \text{TN} \approx |\mathcal{E}_{\text{potential}}|$, which makes AUROC overly optimistic in evaluating model performance. Conversely, AUPRC is better equipped to handle such imbalanced scenarios [57]. As a result, AUPRC can more accurately reflect the performance of the model.

# C Supplementary Contents of AUCell Algorithm

AUCell is designed to quantify the activity of predefined gene regulatory regulons in scRNA-seq data. By calculating the Area Under the Recovery Curve (AUC) for regulons across individual cells, it identifies cells exhibiting coordinated activation of specific transcriptional programs, independent of absolute expression scales.

For a regulon $R$ comprising $m$ genes and a cell $c$ with $n$ detected genes, AUCell operates through three sequential phases. First, genes in cell $c$ are ranked by their expression values in descending order, generating an ordered list $\boldsymbol{g}^c = (g_1^c, g_2^c, \ldots, g_n^c)$, where $g_1^c$ denotes the highest-expressed gene. Ties in expression values are resolved stochastically to avoid rank bias.

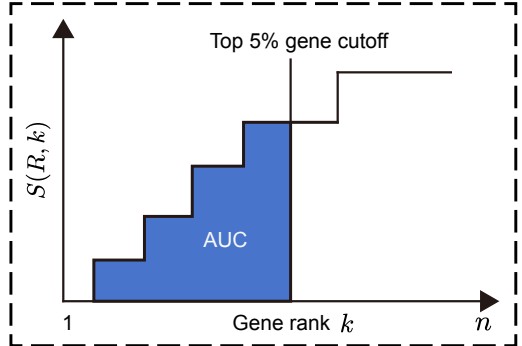

Figure 7: The recovery curve in the AUCell algorithm, with the gene ranking as x-axis and the recovery score as the y-axis.

Subsequently, a binary recovery vector is constructed using an indicator function for regulon membership:

$$\mathbb{I}_R(g_i^c) = \begin{cases} 1, & \text{if } g_i^c \in R \\ 0, & \text{otherwise} \end{cases} \tag{13}$$

The cumulative recovery score $S(R, k)$ is computed by summing $\mathbb{I}_R(g_i^c)$ over the top $k$ genes:

$$S(R, k) = \sum_{i=1}^{k} \mathbb{I}_R(g_i^c) \tag{14}$$

By taking the gene ranking $k$ as the x-axis and the cumulative recovery score $S(R, k)$ as the y-axis, a recovery curve can be plotted, as illustrated in Figure 7. Finally, the AUC score is obtained by calculating the area under this curve. In essence, the AUC score evaluates whether a crucial subset of the input gene set is preferentially enriched among the top-ranked genes in each cell. It also quantifies the proportion of expressed signature genes and their relative expression levels compared to all other genes within the cell, thereby providing a measure of the regulon's activity in that specific cell.

In KINDLE, after calculating the AUC score, we additionally conducted an analysis of variance on this score. Based on statistical significance analysis, we ultimately identified 25 key TFs, with their corresponding P-values summarized in Table 3.

Table 3: P-values and adjusted P-values for selected TFs, with those highlighted in red corresponding to TFs previously reported in the literature.

| Gene | P-value | adjusted P-value | Gene | P-value | adjusted P-value |
|------|---------|------------------|------|---------|------------------|
| Gata4 | $3.8203 \times 10^{-16}$ | $4.4315 \times 10^{-16}$ | Sox17 | $2.4402 \times 10^{-60}$ | $3.9314 \times 10^{-60}$ |
| Kdm5b | $8.1186 \times 10^{-58}$ | $1.2392 \times 10^{-57}$ | Mybl2 | $1.4571 \times 10^{-43}$ | $1.8372 \times 10^{-43}$ |
| Sox2 | $1.2096 \times 10^{-47}$ | $1.6705 \times 10^{-47}$ | Suz12 | $1.9448 \times 10^{-19}$ | $2.3500 \times 10^{-19}$ |
| Nanog | $2.5669 \times 10^{-80}$ | $6.7672 \times 10^{-80}$ | Ets1 | $4.7752 \times 10^{-73}$ | $9.2321 \times 10^{-73}$ |
| Utf1 | $2.4185 \times 10^{-71}$ | $4.3836 \times 10^{-71}$ | Rest | $6.1475 \times 10^{-83}$ | $1.9808 \times 10^{-82}$ |
| Zfp42 | $7.8886 \times 10^{-113}$ | $4.5754 \times 10^{-112}$ | Egr1 | $1.8206 \times 10^{-82}$ | $5.2797 \times 10^{-82}$ |
| Esrrb | $2.3245 \times 10^{-108}$ | $1.1235 \times 10^{-107}$ | Pml | $6.8755 \times 10^{-80}$ | $1.6616 \times 10^{-79}$ |
| Trp53 | $3.8236 \times 10^{-171}$ | $1.1088 \times 10^{-169}$ | Tcf7l2 | $1.2169 \times 10^{-56}$ | $1.7646 \times 10^{-56}$ |
| Sap30 | $3.0508 \times 10^{-119}$ | $4.4236 \times 10^{-118}$ | Trim28 | $1.8347 \times 10^{-105}$ | $7.6010 \times 10^{-105}$ |
| Nr0b1 | $1.2291 \times 10^{-78}$ | $2.7419 \times 10^{-78}$ | Elf1 | $1.9730 \times 10^{-114}$ | $1.4304 \times 10^{-113}$ |
| Nfya | $2.5722 \times 10^{-47}$ | $3.3907 \times 10^{-47}$ | Jarid2 | $5.9858 \times 10^{-68}$ | $1.0211 \times 10^{-67}$ |
| Tcf12 | $9.1998 \times 10^{-119}$ | $8.8931 \times 10^{-118}$ | Bcl3 | $3.2749 \times 10^{-73}$ | $6.7838 \times 10^{-73}$ |
| Akt1 | $1.3803 \times 10^{-90}$ | $5.0038 \times 10^{-90}$ | | | |

# D  Supplementary Contents of In Silico Perturbation

In silico perturbation serves as a critical benchmark for evaluating the accuracy of GRN. By simulating TF perturbation (e.g., knockouts) and propagating their effects through the inferred GRN, this approach quantifies the network's ability to predict downstream gene expression changes and cell fate transitions. The following sections detail the computational framework of CellOracle's in silico perturbation pipeline, which integrates GRN-based signal propagation, cell-state transition modeling, and perturbation score calculation.

## D.1  Signal Propagation for TF Perturbation Simulation

Given an inferred GRN represented by its regulatory coefficient matrix $\mathbf{A} \in \mathbb{R}^{M \times M}$, where $M$ denotes the number of genes. When perturbing a TF $i$, we set its expression to zero. By subtracting the perturbed expression values from the original ones, we obtain a vector $\Delta \mathbf{x}^{(0)} \in \mathbb{R}^{M}$, which is defined as:

$$\Delta x_j^{(0)} = \begin{cases} -x_i, & \text{if } j = i \text{ (TF knockout)} \\ 0, & \text{otherwise} \end{cases} \tag{15}$$

The impact of this perturbation on gene expression is propagated through the matrix $\mathbf{A}$. For the first order perturbation, it is calculated as:

$$\Delta \mathbf{x}^{(1)} = \mathbf{A}^{\top} \Delta \mathbf{x}^{(0)} \tag{16}$$

where $\mathbf{A}^{\top}$, based on the regulatory weights between gene pairs, propagates the perturbation effect to their direct targets. Higher order indirect effects are computed iteratively via $K$ rounds of signal propagation. Specifically:

$$\Delta \mathbf{x}^{(k)} = \mathbf{A}^{\top} \Delta \mathbf{x}^{(k-1)}, \quad k = 2, 3, \ldots, K \ (K = 3 \ for \ defalut) \tag{17}$$

After the $k$-th propagation, the resulting perturbation vector $\Delta \mathbf{x}^{(k)}$ is considered as the simulated perturbation vector $\Delta \mathbf{X}_{\text{sim}}$. It should be noted that during each propagation step, if any element in $\Delta \mathbf{x}^{(k)}$ is less than 0, this element needs to be reassigned as 0, since gene expression levels are always non-negative. Mathematically, this can be expressed as:

$$\Delta \mathbf{x}^{(k)} \leftarrow \max(\Delta \mathbf{x}^{(k)}, 0) \tag{18}$$

## D.2  Cell-State Transition Estimation

The simulated gene expression shifts $\Delta \mathbf{X}_{\text{sim}}$ are translated into cell-state transition probabilities through a kernelized similarity analysis in the two-dimensional embedding space. For each cell $i$, the transition probability $p_{i,j}$ to its $K$-nearest neighbors ($j \in \mathcal{N}_i$) is computed by comparing the simulated perturbation vector $\Delta \mathbf{X}_{\text{sim},i}$ with the observed expression difference $\mathbf{X}_j - \mathbf{X}_i$. This is formalized using a softmax function over Pearson correlation similarities:

$$p_{i,j} = \frac{\exp\left(\rho(\Delta \mathbf{X}_{\text{sim},i}, \mathbf{X}_j - \mathbf{X}_i)/\tau\right)}{\sum_{k \in \mathcal{N}_i} \exp\left(\rho(\Delta \mathbf{X}_{\text{sim},i}, \mathbf{X}_k - \mathbf{X}_i)/\tau\right)} \tag{19}$$

where $\rho$ denotes the Pearson correlation function, $\mathbf{X}_j$ means the expression of gene $j$, $\mathbf{X}_i$ means the expression of gene $i$ and $\tau = 0.05$ modulates the selectivity of the probability distribution. The transition probabilities are then projected onto the two-dimensional embedding space to construct a perturbation vector field. For each cell-neighbor pair, the coordinate difference vector $\mathbf{v}_{i,j} = \mathbf{V}_j - \mathbf{V}_i$ ($\mathbf{V}_i$ means the coordinate of gene $i$ in the two-dimensional space) is weighted by $p_{i,j}$, yielding the simulated perturbation vector for cell $i$:

$$\mathbf{v}_{\text{sim},i} = \sum_{j \in \mathcal{N}_i} p_{i,j} \cdot \mathbf{v}_{i,j} \tag{20}$$

This vector $\mathbf{v}_{\text{sim},i}$ represents the predicted direction and magnitude of cell-state transition induced by the TF perturbation. To account for cellular heterogeneity, this process is repeated across all cells, generating a global perturbation vector field $\mathbf{V}_{\text{sim}} \in \mathbb{R}^{N \times 2}$, where $N$ is the number of cells. This vector field captures context-dependent regulatory effects, enabling systematic visualization of simulated differentiation trajectories.

### D.3 Perturbation Score Calculation

The perturbation score (PS) quantifies the alignment between simulated perturbation-driven cell-state transitions and intrinsic differentiation trajectories. The intrinsic differentiation vector field $\mathbf{V}_{\text{diff}} \in \mathbb{R}^{N \times 2}$ is derived as the spatial gradient of pseudotime $t$, where pseudotime (inferred via diffusion pseudotime or RNA velocity) represents the progression of cells along developmental trajectories. Specifically, $\mathbf{V}_{\text{diff},i} = \nabla t_i$ captures the direction and magnitude of natural differentiation for cell $i$ in the low-dimensional embedding space. To evaluate the impact of TF perturbation, we compute the cosine similarity between the simulated perturbation vector $\mathbf{v}_{\text{sim},i}$ and the differentiation vector $\mathbf{v}_{\text{diff},i}$:

$$\text{PS}_i = \frac{\mathbf{v}_{\text{sim},i} \cdot \mathbf{v}_{\text{diff},i}}{\|\mathbf{v}_{\text{sim},i}\| \|\mathbf{v}_{\text{diff},i}\|} \tag{21}$$

where a positive PS (green in Figure 4c,d) indicates that the perturbation promotes differentiation along the native trajectory, while a negative PS (purple in Figure 4c,d) suggests suppression of differentiation. This directional alignment metric enables systematic identification of TFs that act as drivers or brakes in cell fate determination.

# E Supplementary Contents of Whole TF's AUC Score

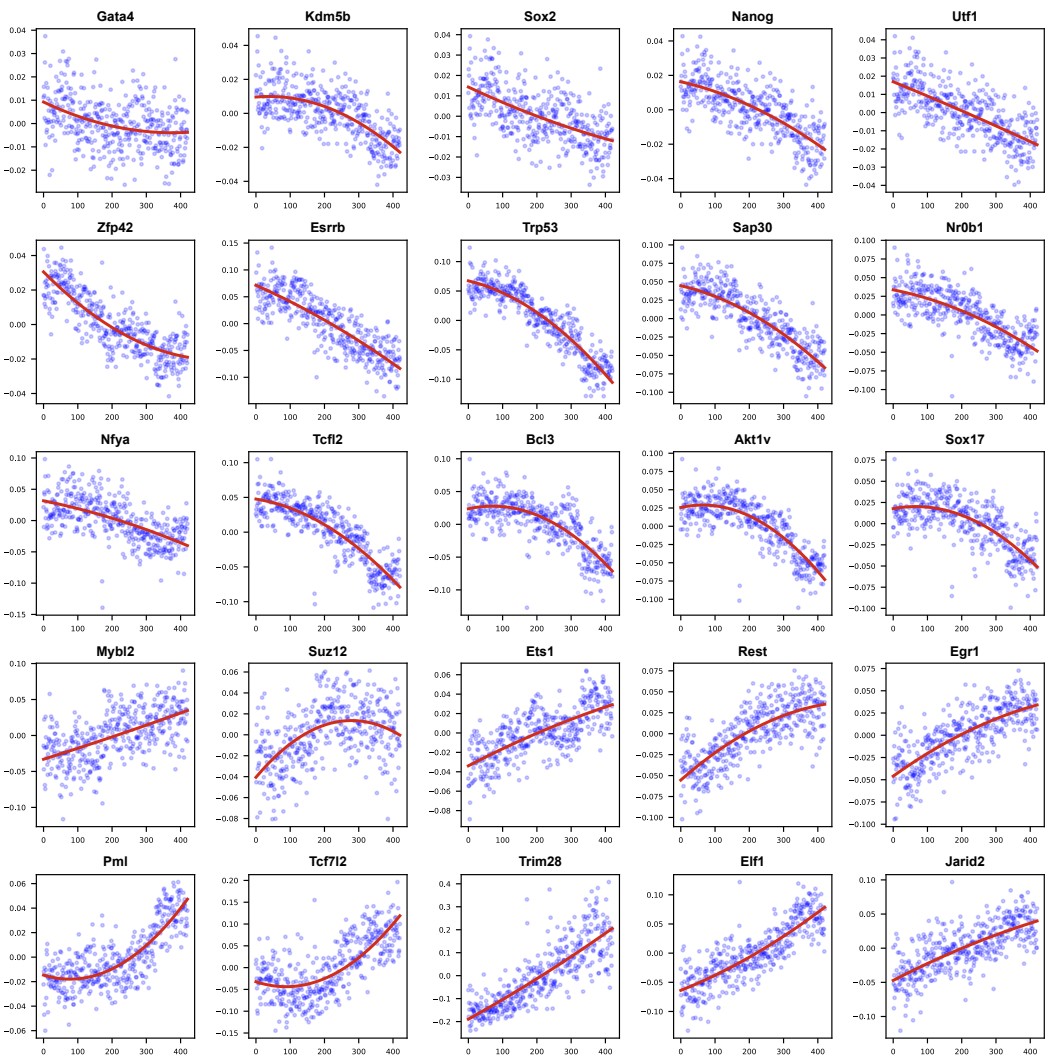

Figure 8: Temporal dynamics of AUC scores for all 25 TFs identified by KINDLE. Blue scatter points represent AUC scores at individual time points, while the red curve denotes a quadratic fitting of these scores.

# F Supplementary Contents of KINDLE Algorithm

**Algorithm 2** KINDLE Framework for GRN Inference

1: **Input**:
2: - Temporal expression matrix: $\mathbf{G} \in \mathbb{R}^{N \times M}$ with $N$ time points and $M$ genes
3: - Spatial prior mask for teacher model: $\mathcal{M}_{spatial} \in \{0,1\}^{M \times M}$
4: **Output**: Inferred GRN adjacency matrix $\mathcal{G}_{pred} \in \mathbb{R}^{M \times M}$
5: // Teacher Training Stage, Initialize Teacher $f_{\theta_T}$ with parameters $\Theta_T$
6: **for** epoch $\in [1, E_{\max}]$ **do**
7:     Sample batch $\mathcal{B} \sim \mathbf{G}$ where $|\mathcal{B}| = B$
8:     **for** each sequence $X_{1:T+W}^{(i)}$ **do**         $\triangleright$ $T$: Historical window, $W$: Prediction window
9:         Partition $X^{(i)} \to (X_{1:T}^{(i)}, Y_{T+1:T+W}^{(i)})$ $\triangleright$ Partition sequence into historical and future parts
10:         $Q_t, K_t, V_t = QKV(X_{1:T}^{(i)})$         $\triangleright$ Get $Q, K, V$
11:         $A_t = \text{softmax}\left(\frac{Q_t K_t^\top}{\sqrt{d_k}} \odot \mathcal{M}_{temporal}\right)$     $\triangleright$ Compute temporal attention
12:         $H_t = A_t V_t \in \mathbb{R}^{B \times T \times M}$         $\triangleright$ Obatin input for spatial layer
13:         $Q_s, K_s, V_s = QKV(\psi(H_t))$         $\triangleright$ $\psi$: Transposition operation
14:         $A_s = \text{softmax}\left(\frac{Q_s K_s^\top}{\sqrt{d_k}} \odot \mathcal{M}_{spatial}\right)$
15:         $\widehat{Y}^{(i)} = \psi(A_s V_s) W_h \in \mathbb{R}^{B \times W \times M}$     $\triangleright$ $W_h$ : Weights for output layer
16:         Update $\Theta_T \propto \nabla_{\Theta_T}\left[\frac{1}{BWM}\sum\left\|\widehat{Y}^{(i)} - Y^{(i)}\right\|_2^2\right]$     $\triangleright$ Update teacher model parameters
17:     **end for**
18: **end for**
19: **return** Teacher model $f_T(\cdot\,;\Theta_T)$
20: // Student Distillation Stage, Initialize Student $f_{\theta_S}$ with parameters $\Theta_S$ and frozen Teacher $f_{\theta_T}$
21: **for** epoch $\in [1, E_{\text{distill}}]$ **do**
22:     **for** each $X_{1:T+W}^{(i)} \in \mathcal{B}$ **do**
23:         Partition $X^{(i)} \to (X_{1:T}^{(i)}, Y_{T+1:T+W}^{(i)})$
24:         $\widehat{Y}_T^{(i)} = f_T(X_{1:T}^{(i)})$         $\triangleright$ Get teacher predictions
25:         $\widehat{Y}_S^{(i)} = f_S(X_{1:T}^{(i)})$         $\triangleright$ Get student predictions
26:         $\mathcal{L}_{\text{pred}} = \frac{1}{BWM}\sum\|\widehat{Y}_S^{(i)} - Y^{(i)}\|^2$     $\triangleright$ Compute student prediction loss
27:         **if** Distillation Type = "Hard" **then**     $\triangleright$ Choose distillation loss
28:             $\mathcal{L}_{\text{distill}} = \frac{1}{BWM}\sum\|\widehat{Y}_T^{(i)} - \widehat{Y}_S^{(i)}\|_2^2$
29:         **else if** Distillation Type = "Soft" **then**
30:             $\mathcal{L}_{\text{distill}} = \frac{1}{BWM}\sum\text{KL}\left(\sigma\left(\frac{\widehat{Y}_T^{(i)}}{\tau}\right)\,\middle\|\,\sigma\left(\frac{\widehat{Y}_S^{(i)}}{\tau}\right)\right)$     $\triangleright$ $\tau > 0$: Softmax temperature
31:         **else if** Distillation Type = "Bilinear" **then**
32:             $\mathcal{L}_{\text{distill}} = \frac{1}{BWM}\sum(\widehat{Y}_T^{(i)})^\top(\widehat{Y}_S^{(i)})$
33:         **else if** Distillation Type = "Gaussian" **then**
34:             $\mathcal{L}_{\text{distill}} = \frac{1}{BWM}\sum\exp\left(-\frac{\|\widehat{Y}_T^{(i)} - \widehat{Y}_S^{(i)}\|_2^2}{2\lambda^2}\right)$
35:         **end if**
36:         $\mathcal{L}_{\text{total}} = \alpha\mathcal{L}_{\text{pred}} + (1-\alpha)\mathcal{L}_{\text{distill}}$     $\triangleright$ $\alpha \in [0,1]$: Knowledge distillation coefficient
37:         Update $\Theta_S \propto \nabla_{\Theta_S}\mathcal{L}_{\text{total}}$     $\triangleright$ Update student model parameters
38:     **end for**
39: **end for**
40: **return** Student model $f_S(\cdot\,;\Theta_S)$
41: // GRN Inference Stage
42: Partition $\mathbf{G}$ into $H$ subsequences $\{\mathcal{S}^{(h)}\}_{h=1}^H$ with stride $T$
43: Initialize adjacency confidence matrix $\bar{A} = \mathbf{0}^{M \times M}$
44: **for** each $\mathcal{S}^{(h)} \in \{\mathcal{S}^{(h)}\}_{h=1}^H$ **do**
45:     $\bar{A} \leftarrow \bar{A} + \frac{1}{H}\phi(f_S(\mathcal{S}^{(h)}))$     $\triangleright$ $\phi$: Extract attention matrix in student's spatial layer
46: **end for**
47: **return** $\mathcal{G}_{pred} = \text{top}_k(\bar{A})$     $\triangleright$ Select top k edges

