# OpenReview forum: "KINDLE: Knowledge-Guided Distillation for Prior-Free Gene Regulatory Network Inference"
_NeurIPS.cc/2025/Conference — NeurIPS 2025 poster_

### Official Review · Reviewer_foG6 · 2025-06-30

**Clarity:** 2
**Significance:** 3
**Originality:** 2
**Rating:** 4
**Confidence:** 3

**Summary:**

The paper proposed a method called KINDLE for gene regulatory network (GRN) inference. KINDLE solves the limitation of relying on biologically plausible gene-gene interactions observed in prior knowledge. KINDLE distills encoded prior knowledge by a Teacher model into the Student model not to rely on any prior.

**Questions:**

Please see the Weaknesses part and the following additional questions:

1. What happens if we set T=1 and W=1 in the equations and some of the experiments? How does this affect the output and efficiency of the model?
2. In what manner does KINDLE manage to detect gene interactions that are not included in the prior knowledge dataset?
3. Could you elaborate on the process of chunking the T time steps? What criteria or methodology are used for this chunking?
4. How many GPU hours were required for training KINDLE? How does the computational cost of KINDLE compare to other baseline methods in terms of efficiency and resource consumption?
5. A visualization of which regulatory insights the teacher transfers to the student would be helpful.

**Ethical Concerns:**

["NO or VERY MINOR ethics concerns only"]

**Final Justification:**

The authors have notably addressed the majority of my concerns, especially those related to new interactions. However, the extra analysis and ablation studies I anticipated were absent in their rebuttal, although the training time is reasonable to run additional experiments during the rebuttal week.
I will keep my current rating score, 4 - Borderline Accept.

**Limitations:**

Please see weaknesses.

**Quality:**

3

**Strengths And Weaknesses:**

**Strengths**:
- The student model achieves SOTA performance, demonstrating that causal regulatory knowledge is effectively compressed into a simpler architecture.
- Temporal ensembling of attention matrices (Eq. 11) smooths stochastic noise in single-cell data, improving edge-calling reliability.

**Weaknesses**:

1. "W" is an important variable to evaluate the causality aspect of prediction, but was not explicitly specified in experiments. An ablation on W would really help to compare causal prediction ability to baselines.
3. Gaussian RBF distillation outperforms other losses but requires careful tuning of λ (kernel width). No analysis shows robustness to λ variation.

---

> ### Author Rebuttal · Authors · 2025-07-28
>
> **We sincerely appreciate your thoughtful and valuable feedback. We will address the concerns and questions point by point in the following responses.**
>
> ---
>
> ### ***Question 1***
> "W" is an important variable to evaluate the causality aspect of prediction, but was not explicitly specified in experiments. An ablation on W would really help to compare causal prediction ability to baselines.
>
> ### ***Our response***
> We apologize for the oversight of not explicitly specifying the variable $W$ in the experiments. During the KINDLE training process, we conducted experiments with five distinct values of $W$ (1, 2, 4, 8, and 16). Among these, $W = 16$ yielded the optimal results, which are the results reported in the paper. We will add a clarification regarding the setting of $W$ in Lines 291-296 to enhance reader comprehension.
>
> ---
>
> ### ***Question 2***
> Gaussian RBF distillation outperforms other losses but requires careful tuning of λ (kernel width). No analysis shows robustness to λ variation.
>
> ### ***Our response***
> We wish to emphasize that prior to applying the Gaussian RBF kernel, the input features underwent standardization ($\mu = 0$, $\sigma = 1$). **This preprocessing step substantially mitigates the model's sensitivity to $\lambda$ by ensuring distances in the feature space are on comparable scales**. We will add clarification regarding the $\lambda$ setting in Lines 291-296 to enhance reader comprehension.
>
> ---
>
> ### ***Question 3***
> What happens if we set $T=1$ and $W=1$ in the equations and some of the experiments? How does this affect the output and efficiency of the model?
>
> ### ***Our response***
> Setting both $T$ and $W$ to 1 **enables the inference of a GRN for each individual cell**, thereby capturing single-cell resolution dynamic GRNs. Specifically, since each cell is assigned a pseudotime, each time step corresponds to a single cell. Under the configuration where $T = 1$ and $W = 1$, KINDLE takes the gene expression profile of the cell at the preceding pseudotime step as input and predicts the expression profile of the subsequent cell. Consequently, the model effectively learns the GRN associated with the preceding cell. **This approach allows us to infer the GRN for every cell along the developmental trajectory**. **By temporally aligning the GRNs inferred for each cell, we can observe the dynamic evolution of GRNs throughout development**. For instance, focusing on the regulatory interaction between Gene A and Gene B, their inferred regulation strength within each cell-specific GRN varies. This enables the reconstruction of the temporal trajectory of the regulatory strength for this gene pair, allowing observation of up- or down-regulation at specific developmental phases, thereby yielding deeper mechanistic insights into underlying molecular processes.
>
> As noted in our response to Question 1, we evaluated five distinct values for $W$ (1, 2, 4, 8, and 16). However, we observed that the performance achieved with $T = 1$ and $W = 1$ was inferior to that obtained when both parameters were set to 16. We attribute this primarily to technical limitations inherent in single-cell RNA sequencing, which introduce significant noise and bias into the gene expression data. A key challenge is data sparsity, often manifested as dropout events (genes with actual expression that are erroneously recorded as zero counts due to insufficient sequencing depth). **When $T$ and $W$ are both set to 1, this noise and sparsity disproportionately impair model performance**. In contrast, increasing the number of input cells processed by the model (e.g., setting $T$ and $W$ to 16) mitigates these effects by effectively averaging out stochastic noise. This leads to enhanced model robustness and optimal performance. We will provide a more detailed explanation regarding the impact of $T$ and $W$ settings on model behavior in Lines 297-315 to improve reader comprehension.
>
> ---
>
> ### ***Question 4***
> In what manner does KINDLE manage to detect gene interactions that are not included in the prior knowledge dataset?
>
> ### ***Our response***
> We precisely employ privileged-feature distillation to ensure KINDLE detects gene interactions beyond those included in the prior knowledge. Specifically, prior-based methods constrain the model to learn only within the $K$ edges provided by the prior knowledge, precluding the inference of gene interactions out of prior. To leverage the prior knowledge while overcoming this fundamental limitation, we adopt a distillation strategy. The teacher model incorporates the prior network as input, which acts as a mask within the Transformer's self-attention mechanism. This mask restricts attention computation exclusively to the $K$ edges specified by the prior knowledge. The knowledge learned by the teacher model from the prior is then distilled into the student model. Critically, the student model does not receive the prior knowledge as input. Instead, it learns features transferred from the teacher model. Furthermore, **the self-attention computation within the student model operates without any masking mechanism** (refer to the difference in the spatial layers between the teacher and student models in Figure 2). **This enables the student model to compute interactions between all possible gene pairs rather than be restricted in the $K$ edges from prior**. Thus, while learning the features distilled from the teacher, the student model simultaneously retains the capacity to detect gene interactions absent from the prior knowledge, thereby achieving high inference accuracy and resolving the inherent limitation of prior-based methods.
>
> ---
>
> ### ***Question 5***
> Could you elaborate on the process of chunking the T time steps? What criteria or methodology are used for this chunking?
>
> ### ***Our response***
> We apologize for the omission of a detailed description regarding the process of chunking the $T$ time steps. We will now provide a comprehensive explanation of this procedure. let $G \in \mathbb{R}^{N \times M}$ denotes the temporal single-cell expression matrix, where $N$ represents temporally ordered cellular states and $M$ is the number of genes. our time series data $G = [\mathbf{e}_1,\ldots,\mathbf{e}_N]^T$ possesses the temporal relationships of genes during the differentiation process as we map each cell to a pseudotime (**each cell corresponds to a single time step**), $\mathbf{e}_i \in \mathbb{R}^{M}$ represents the gene expression at the $i$-th time step. After that, we split $G$ into dataset $\mathcal{D} = ((\mathbf{X}_i,\mathbf{Y}_i)|i = 1,\ldots,L)$, where $\mathbf{X}_i$ are input samples, $\mathbf{Y}_i$ are the ground truth associated to each sample and $L$ is the number of samples. This yields the assignments that $\mathbf{X}_i \coloneqq G[i:i+T,:]$ and $\mathbf{Y}_i \coloneqq G[i+T:i+T+W,:]$, where $G[a:b,:]$ denotes the row submatrix from index $a$ to $b-1$, capturing $T$ and $W$ consecutive cells respectively. **In conclusion, we construct input $\mathbf{X}$ and ground truth $\mathbf{Y}$ through sliding-window sampling on the temporally ordered cell matrix $G$, utilizing adjacent windows of length $T$ and $W$ respectively**.
>
> ---
>
> ### ***Question 6***
> How many GPU hours were required for training KINDLE? How does the computational cost of KINDLE compare to other baseline methods in terms of efficiency and resource consumption?
>
> ### ***Our response***
> When $T$ and $W$ are set to 16, KINDLE requires an average of 20 minutes to complete 30 training epochs across all four datasets on an 80GB NVIDIA A100 GPU, with total GPU memory consumption of 30GB. **KINDLE is more efficient compared to prior-based baselines CEFCON and Celloracle, while maintaining comparable runtime to NetREX**. Specifically, CEFCON employs a graph neural network architecture that incurs higher computational complexity than KINDLE's Transformer framework and the requirement to compute both a real network and a false network for contrastive learning substantially increases its training time (approximately 1 hour). Celloracle's execution time is often prolonged due to the computationally intensive process of constructing prior networks from scATAC-seq data, which can take several hours to days. Additionally, its GRN inference relies on iterating through each gene to perform ridge regression, resulting in low efficiency.
>
> **Regarding memory usage, KINDLE exhibits comparable but marginally higher GPU memory consumption than the other methods**. This is attributable to the simultaneous loading and computation of both teacher and student models during training. We acknowledge this limitation entails a risk of memory overflow when processing large-scale datasets. To mitigate this, we plan to explore memory-optimized attention implementations (e.g., Flash Attention) in future optimization of KINDLE.
>
> ---
>
> ### ***Question 7***
> A visualization of which regulatory insights the teacher transfers to the student would be helpful.
>
> ### ***Our response***
> We agree that visualizing the features transferred from the teacher to the student model would enhance reader comprehension. However, under the current official regulations, we are not permitted to upload experimental results such as images or videos. We commit to making these visualizations available once permission is granted.
>
> ---
>
> **We hope the responses provided above adequately address the reviewer's concerns. If there are any aspects that remain unclear, please do not hesitate to contact us further, we remain fully available to provide additional details.**

---

> > ### Comment · Reviewer_foG6 · 2025-07-31
> >
> > Thank you for your response and for providing additional information.
> >
> > Regarding questions 4 and 7, it is still unclear to me about the "features" to be transferred that help to explore new interactions.
> >
> > 1. Could you verbally describe how to visualize and what to expect to see in the plot for question 7?
> > 2. Are the transferred information components latent variables, or are they some interpretable features?
> >
> > Thank you.

---

> ### Author Response · Authors · 2025-08-01
>
> We appreciate the reviewer's insightful questions. We will address them below by clarifying the nature of the feature and describing the visualization approach.
>
> ---
>
> ### 1. **What is the feature transferred from the teacher to the student?**
> As formalized in Equation (4), the distillation loss constrains the relationship between the teacher's output $f\_{\theta\_{T}}(\mathbf{G}\_{1:T})$ and the student's output $f\_{\theta\_{S}}(\mathbf{G}\_{1:T})$. Teacher model incorporates a spatial layer that computes an $M×M$ attention matrix ($M$ is the number of genes). This matrix quantifies attention scores between gene pairs, representing regulatory strength within the GRN context. During self-attention computation, this matrix is multiplied by the value tensor $V$ to generate the final outputs $f\_{\theta\_{T}}(\mathbf{G}\_{1:T})$ (see pseudocode, page 21, lines 13-15 for detailed information). **This attention matrix is referred to the transferred feature**. Crucially, within the teacher model, this matrix integrates prior knowledge via a masking mechanism, enabling the learning of comprehensive gene-gene regulatory relationships guided by this prior. **It thereby encodes inter-gene dependencies and directs the model’s predictions of future gene expression values**. Calculation of the distillation loss between $f\_{\theta\_{T}}(\mathbf{G}\_{1:T})$ and $f\_{\theta\_{S}}(\mathbf{G}\_{1:T})$ indirectly transfers this feature (**specifically, the prior-informed attention patterns**) from teacher to student.
>
> ### 2. **How to visualize the feature transfer?**
> To investigate how the feature transferred from the teacher influences the student, we extract the attention matrix $A\_{T}$ and $A\_{S}$ from the spatial layer of teacher and student model, and employ the following visualization methods:
> - **Heatmap**: We generate separate heatmaps for $A\_{T}$, $A\_{S}$, and their difference matrix $A\_{S} - A\_{T}$. Visual inspection reveals that $A\_{T}$ and $A\_{S}$ exhibit broadly similar patterns. The difference heatmap shows extensive regions of minimal deviation (visualized as white), **indicating successful transfer of the teacher's prior-informed attention patterns to the student model**. Conversely, localized areas of higher deviation (visualized as red) reflect regulatory relationships learned independently by the student, which lie outside the scope of the prior knowledge.
> - **Scatter Plot**: We extract the corresponding attention scores from both matrices and generate a scatter plot with $A\_{T}$ on the x-axis and $A\_{S}$ on the y-axis. Each point represents the attention score for a specific gene pair $(i, j)$. This visualization reveals that a large majority of points cluster tightly around $y = x$. This demonstrates that the student has effectively learned the regulatory pattern for these gene pairs directly from the teacher. Points deviating significantly from $y = x$ represent regulatory relationships learned independently by the student, not covered by the prior knowledge.
>
> ---
>
> We sincerely appreciate your insightful feedback, which has significantly strengthened our work. We hope the responses provided comprehensively address the reviewer's concerns. Should any aspects require further clarification, we remain available to provide additional details upon request. If you find we have successfully addressed your concerns, we would be grateful for your reconsideration of our manuscript. Thanks a lot!

---

> > ### Comment · Reviewer_foG6 · 2025-08-04
> >
> > Thank you for providing more information about the transferred information and the difference plots. I appreciate your effort in addressing my concerns.
> >
> > I have some follow-up questions:
> >
> > 1. Is the attention mask context-dependent? For example, can the potential for gene-gene interactions be affected by certain specific conditions?
> >
> > 2. If the mask \( A_T \) reflects the gene-gene interactions trained from prior knowledge, and \( A_T \) is _expected_ to transfer to \( A_S \) with an exact 1-1 mapping, how can the student use \( A_S \) to explore knowledge beyond what was used in training \( A_T \), as claimed in the paper? My understanding is that we can potentially explore new concepts (new interactions) by decomposing knowledge into pieces, and then combining these pieces to form new concepts (new combinations of knowledge that go beyond those in the priors). Could you please help clarify this point?
> >
> > 3. In the heatmap description with the difference matrix \( A_S - A_T \), I believe that cases where \( A_S \) identifies interactions not present in \( A_T \) are unavoidable, particularly when the value of \( A_T \) is zero or small, while \( A_S \) is positive or significantly higher. Have you observed these patterns in the map? Additionally, do you have any explanations or intuitions about why this occurs and how it might affect the analysis?
> >
> > Thank you!

---

> > > ### Author Response · Authors · 2025-08-05
> > >
> > > We appreciate the reviewer's insightful questions and will address these one by one.
> > >
> > > ---
> > >
> > > ### ***Question 1***
> > > Is the attention mask context-dependent? For example, can the potential for gene-gene interactions be affected by certain specific conditions?
> > >
> > > ### ***Our response***
> > > The attention mask is context-dependent and linked to cellular developmental processes. Specifically, consider cells originating from the same cell type A. When one subset differentiates into cell type B while another differentiates into cell type C, KINDLE infers distinct attention mask for the $A \rightarrow B$ versus the $A \rightarrow C$ developmental trajectories. This divergence in attention masks is critical, **it enables the identification of specific key genes that govern differentiation towards lineage B rather than lineage C**, and provides deeper insights into the underlying regulatory mechanisms driving cell fate decisions.
> > >
> > > ---
> > >
> > > ### ***Question 2***
> > > If the mask ( A_T ) reflects the gene-gene interactions trained from prior knowledge, and ( A_T ) is expected to transfer to ( A_S ) with an exact 1-1 mapping, how can the student use ( A_S ) to explore knowledge beyond what was used in training ( A_T ), as claimed in the paper? My understanding is that we can potentially explore new concepts (new interactions) by decomposing knowledge into pieces, and then combining these pieces to form new concepts (new combinations of knowledge that go beyond those in the priors). Could you please help clarify this point?
> > >
> > > ### ***Our response***
> > > The student model infers new interactions via an edge prediction-like approach. As detailed in **Lines 146-148**, the student model's attention computation operates without any masking mechanism, ensuring that attention scores are calculated for all possible gene pairs. Specifically, the teacher model transfers the prior knowledge network to the student model via knowledge distillation. Upon receiving this directed graph, the student model leverages it as a foundation to compute regulatory scores for edges not present within this prior knowledge. Edges ultimately exhibiting sufficiently high predicted scores are incorporated into the inferred GRN. **This process fundamentally constitutes edge prediction on a graph, after learning the prior knowledge, the model predicts which novel edges are likely to represent regulatory relationships**. This aligns with the reviewer's conceptualization that the student model effectively decomposes knowledge into pieces and, during the process of combining these pieces, computes regulatory scores for potential new edges. Those edges attaining high regulatory scores are subsequently identified as new interactions.
> > >
> > > ---
> > >
> > > ### ***Question 3***
> > > In the heatmap description with the difference matrix ( A_S - A_T ), I believe that cases where ( A_S ) identifies interactions not present in ( A_T ) are unavoidable, particularly when the value of ( A_T ) is zero or small, while ( A_S ) is positive or significantly higher. Have you observed these patterns in the map? Additionally, do you have any explanations or intuitions about why this occurs and how it might affect the analysis?
> > >
> > > ### ***Our response***
> > > Indeed, as the reviewer notes, the presence of zero or small values in $A\_T$ inevitably leads the student model $A\_S$ to identify interactions absent from $A\_T$. **However, we wish to emphasize that not every edge exhibiting a difference from $A\_T$ is classified as a new interaction**. As detailed in our response to Question 2, our approach involves calculating a regulatory score for each candidate new edge. Crucially, only edges attaining high regulatory scores are identified as new interactions. This selectivity arises because the final GRN is constructed by selecting the top $k$ edges based on their predicted scores. **Consequently, edges absent from $A\_T$ but assigned low regulatory scores by the student model are excluded from the inferred GRN**. This is corroborated by our heatmap analysis that the vast majority of edges not present in $A\_T$ exhibit low computed regulatory scores, consequently appearing white (indicating negligible difference) in the difference map. In contrast, novel interactions predicted by the student model with high regulatory scores manifest as distinct red in the difference map. Importantly, subsequent validation against ground truth confirmed that these high-scoring, student-identified new interactions are correct.
> > >
> > > ---
> > >
> > > We hope the responses provided above adequately address the reviewer's concerns. If there are any aspects that remain unclear, please do not hesitate to contact us further, we remain fully available to provide additional details.

---

> > > > ### Comment · Reviewer_foG6 · 2025-08-06
> > > >
> > > > Thank you for your explanation.
> > > >
> > > > I want to express my appreciation for your efforts in addressing my questions and follow-up points during the discussion period.
> > > >
> > > > The authors have done a commendable job in responding to most of my concerns, particularly regarding the new interactions. However, I noticed that the additional analysis and ablation studies I was expecting were not provided in the rebuttal. I recommend including the necessary analyses and explanations in the revised version to further strengthen the paper.
> > > >
> > > > I will maintain my current rating score.

---

### Official Review · Reviewer_2Z9D · 2025-07-02

**Clarity:** 4
**Significance:** 4
**Originality:** 4
**Rating:** 6
**Confidence:** 4

**Summary:**

This paper introduces KINDLE, a framework designed to enhance the accuracy of GRN inference by eliminating fixed prior network information. The authors test this three-stage framework on both the mouse embryonic stem cell and the mouse hematopoietic stem cell development.

**Questions:**

Can you provide reference(s) for the claim made on page 1, lines 31 -33 regarding the quadratic scaling?

**Ethical Concerns:**

["NO or VERY MINOR ethics concerns only"]

**Limitations:**

yes

**Quality:**

4

**Strengths And Weaknesses:**

The paper clearly explains details and makes it easy to follow along.

---

> ### Author Rebuttal · Authors · 2025-07-29
>
> We are deeply grateful to the reviewer for your recognition of our manuscript. We have carefully read your thoughtful and valuable feedback and will provide point-by-point responses to address your concerns and questions.
>
> Regarding the references for quadratic scaling in lines 31-33, **we primarily drew upon BEELINE [1] and a comprehensive review on GRN inference [2]**. BEELINE establishes a benchmark framework for GRN evaluation and proposes various metrics to assess the accuracy, robustness, and computational efficiency of GRN inference methods. The review article provides a systematic examination of the field, encompassing single-modal and multi-modal inference approaches, downstream GRN analyses, as well as current limitations and future research directions.
>
> Both references offer authoritative summaries of the GRN inference landscape, delivering valuable insights into methodological approaches, persistent challenges, and evaluation strategies within the field. To strengthen reader comprehension, we will explicitly add citations for these two articles in lines 31-33.
>
> [1] Aditya Pratapa, Amogh P Jalihal, Jeffrey N Law, Aditya Bharadwaj, and TM Murali. Benchmarking algorithms for gene regulatory network inference from single-cell transcriptomic data. Nature methods, 17(2):147–154, 2020.
>
> [2] Pau Badia-i Mompel, Lorna Wessels, Sophia Müller-Dott, Rémi Trimbour, Ricardo O Ramirez Flores, Ricard Argelaguet, and Julio Saez-Rodriguez. Gene regulatory network inference in the era of single-cell multi-omics. Nature Reviews Genetics, 24(11):739–754, 2023.
>
> ---
>
> **We hope the responses provided above adequately address the reviewer's concerns. If there are any aspects that remain unclear, please do not hesitate to contact us further, we remain fully available to provide additional details**.

---

### Official Review · Reviewer_7DWh · 2025-07-02

**Clarity:** 4
**Significance:** 3
**Originality:** 4
**Rating:** 5
**Confidence:** 4

**Summary:**

The authors proposed a deep learning framework, KINDLE, to infer gene regulatory network (GRN) using teacher -student modeling and knowledge distillation with gene expression information, temporal gene expression profiles and known gene-gene regulation. During test phase on new data, only gene expression profile is needed. The four variants of the proposed approach achieve the highest performance in terms of area under the receiver operating characteristic curve (AUROC), area under the precision-recall curve (AUPRC), and F1 score on four data sets: Mouse embryonic stem cell (mESC), Mouse embryonic stem cell (mESC), Erythrocyte (mHSC-E), Granulocyte-Monocyte (mHSC-GM), and Lymphocyte (mHSC-L), compared with baseline methods that utilize or do not utilize prior knowledge.

**Questions:**

1. How does prior information help in the framework? Or what process is to be adapted to minimize the potential bias distillation process?
2. In common biological experiments, there are some conditions with a few replicates for each condition. Please discuss the potential adoption of the current framework to this type of data.
3. There are more improvement in terms of AUPRC, compared with AUROC. Can the authors explain what could be the reason?

**Ethical Concerns:**

["NO or VERY MINOR ethics concerns only"]

**Final Justification:**

The authors answered question on what process is to be adapted to minimize the potential bias distillation process, on scenario with a few replicates for each condition, and on more improvement in AUPRC.

**Limitations:**

yes

**Quality:**

4

**Strengths And Weaknesses:**

The submission is technically sound.  The theoretical analysis and experimental results are well supporting the authors' claim. The methods used are appropriate. The work is almost complete piece of work. The authors mentioned both strengths and weaknesses of their work carefully and honestly.

The submission is clearly written and well organized. Some of the details may need more explanation. No source code was provided.

The authors use teacher - student model and knowledge distillation to improve gene regulatory network (GRN) inference. The framework achieves the highest performance comparing with some of the state of the art methods, given GRN inference is a difficult task. It is likely that when temporary gene expression data are available for the cell type of interest, the proposed approach will be used.

The work proves that teacher - student model and knowledge distillation improve GRN inference on the datasets tested, showing the importance of knowledge distillation. The authors clearly distinguish their contributions with previous contributions. The authors offer a novel combination of existing techniques, and the reasoning behind this combination is well-articulated.

---

> ### Author Rebuttal · Authors · 2025-07-29
>
> **We are deeply grateful to the reviewer for your recognition of our manuscript. We have carefully read your thoughtful and valuable feedback, and will provide point-by-point responses below to address your concerns and questions.**
>
> ---
>
> ### ***Question 1***
> How does prior information help in the framework? Or what process is to be adapted to minimize the potential bias distillation process?
>
> ### ***Our response***
> 1. The prior information utilized in our framework is compiled from over 50 public data sources, consolidating regulatory pairs extracted from extensive experimental studies and literature. Within the transformer architecture, this prior functions as an attention mask during self-attention computation. It explicitly constrains the attention mechanism to focus solely on edges documented in the prior knowledge, while suppressing computation for irrelevant gene pairs. **Through this guided attention mechanism, the teacher model develops a global understanding of gene regulatory relationships**. The features learned under this prior guidance are subsequently distilled into the student model, enabling the latter to maintain high predictive performance without explicit prior constraints during inference.
> 2. During experimentation, we observed that gene expression data exhibits inherent sparsity primarily attributable to technical limitations of sequencing protocols (e.g., dropout events where expressed genes yield zero counts). **This sparsity introduces systematic biases in raw expression values**. As both teacher and student models take these expression profiles as input during distillation, the technical biases become progressively amplified throughout the knowledge transfer process, ultimately compromising model performance. To mitigate this potential bias propagation, **we propose applying imputation algorithms as a preprocessing step**. These methods estimate biologically plausible expression values for technical zeros, thereby recovering expression levels obscured by sequencing artifacts, reducing systematic bias in input data, and ultimately enhancing the overall accuracy of the distillation process.
>
> ---
>
> ### ***Question 2***
> In common biological experiments, there are some conditions with a few replicates for each condition. Please discuss the potential adoption of the current framework to this type of data.
>
> ### ***Our response***
> Thank you for raising this important point regarding the applicability of KINDLE to datasets with limited replicates, which is indeed a common scenario in biological experiments. We propose that KINDLE can adapt to such data through three potential solutions:
> ### **1. Take advantage of consolidated prior knowledge**
> The teacher model is trained on a consolidated prior network integrating experimentally validated regulatory relationships sourced from over 50 public databases and literature resources. Critically, these foundational datasets originate from large-scale experimental studies **inherently possessing substantial replicate samples**. When inferring GRNs for a target condition with sparse replicates, the scale and biological relevance of this integrated prior knowledge become paramount. By distilling patterns learned from these evidence-rich, **high-replicate sources**, the teacher model provides the student model with a robust foundation of general regulatory principles. This significantly reduces the student model's reliance on extensive replicates, effectively serving as a powerful biologically grounded regularizer to compensate for data sparsity in the target condition.
>
> ### **2. Leverage appropriate data augmentation strategies**
> 1. **Standard augmentation**: Applying standard data augmentation techniques to the limited expression profiles, such as adding biologically plausible noise.
> 2. **Generative model-based augmentation**: Utilizing generative models (e.g., VAE, GAN and diffusion models) trained on relevant expression data to synthesize plausible additional expression profiles.
>
> It should be noted that employing data augmentation requires careful validation to ensure biological fidelity and avoid introducing artifacts. Nevertheless, these strategies represent potential avenues within the KINDLE framework to further mitigate limitations imposed by scarce replicates.
>
> ### **3. Transfer knowledge from biologically similar cell types**
> During teacher model training, data from biologically similar cell types exhibiting abundant replicate samples can be utilized. Knowledge distilled from these related, replicate-rich datasets is encoded within the teacher model and subsequently transferred to the student model. The student model then undergoes lightweight finetuning using the limited target-condition replicates. **This process mirrors the "pretraining & finetuning" paradigm**, enabling effective adaptation of the downstream model to specific data-scarce scenarios.
>
> ---
>
> ### ***Question 3***
> There are more improvement in terms of AUPRC, compared with AUROC. Can the authors explain what could be the reason?
>
> ### ***Our response***
> We thank the reviewer for raising this point. We wish to emphasize that in **Lines 592-602**, we discuss why AUPRC is a more robust metric compared to AUROC. Specifically, let $\mathcal{E}\_{potential}$ represents the set of all explorable gene pairs and $\mathcal{E}\_{true}$ denotes the edges present in the ground truth network. As detailed in Table 2, which presents the distribution of positive and negative labels across the four datasets, we observe severe label imbalance in all datasets, with negative labels exceeding 99% of the total. In such scenarios characterized by extreme class imbalance, AUROC disproportionately emphasizes the majority class (negative edges). This occurs because AUROC relies heavily on the false positive rate ($FPR = \frac{FP}{(FP + TN)}$). When $\mathcal{E}\_{true} \leq \mathcal{E}\_{potential}$, the sum $FP + TN \approx |\mathcal{E}\_{potential}|$, rendering AUROC overly optimistic in evaluating model performance. Conversely, as analyzed in the literature (i.e., *The Relationship Between Precision-Recall and ROC Curves*), AUPRC is better equipped to handle such imbalanced scenarios. Therefore, AUPRC serves as a more robust metric for assessing model performance under severe label imbalance than AUROC.
>
> Consequently, the significant improvements demonstrated by KINDLE in AUPRC (compared to the more modest gains in AUROC) provide stronger evidence for its superior performance over the baseline models.
>
> ---
>
> **We hope the responses provided above adequately address the reviewer's concerns. If there are any aspects that remain unclear, please do not hesitate to contact us further, we remain fully available to provide additional details**.

---

> > ### Comment · Reviewer_7DWh · 2025-08-08
> >
> > Thanks authors for your detailed response. Your proposals sound promising. For Question 3, AUPRC is preferable in this scenario. Can you elaborate what step(s) of the method help improve the AUPRC?

---

> > > ### Author Response · Authors · 2025-08-08
> > >
> > > Thank you for raising this point. In the computation of the AUPRC, *Precision* and *Recall* are critical metrics, defined as: $Precision=\frac{TP}{TP+FP}$ and $Recall=\frac{TP}{TP+FN}$. The KINDLE enhances AUPRC by optimizing both metrics through its unique methodology. Specifically, KINDLE achieves this via:
> > > 1. **Enhancing Precision through Prior Knowledge Integration**: During teacher model training, KINDLE incorporates prior knowledge. This mechanism guides the model to prioritize biologically validated regulatory relationships within the vast space of potential gene pairs, effectively reducing false positives (FP) by minimizing misidentification of non-regulatory edges as regulatory interactions. Consequently, Precision is improved.
> > > 2. **Increasing Recall via Knowledge Distillation**: The student model learns regulatory pattern recognition from the teacher through knowledge distillation, without being constrained by prior knowledge. This enables exploration across the entire gene pair space, identifying both known regulatory edges and novel, undocumented interactions. Thus, edges falsely labeled as negatives due to prior knowledge exclusion are correctly reclassified as positives by the student model, thereby reducing false negatives (FN) and consequently increasing Recall.
> > >
> > > In summary, KINDLE leverages the teacher model’s prior knowledge-guided refinement to achieve high Precision, while the student model’s prior knowledge-free exploration extends coverage to maximize Recall. The two mechanisms collectively improve AUPRC performance.
> > >
> > > ---
> > >
> > > We hope the responses provided above adequately address the reviewer's concerns. If there are any aspects that remain unclear, please do not hesitate to contact us further, we remain fully available to provide additional details.

---

### Official Review · Reviewer_UYaC · 2025-07-02

**Clarity:** 2
**Significance:** 1
**Originality:** 3
**Rating:** 4
**Confidence:** 3

**Summary:**

The paper presents a novel framework for learning the adjacency matrix of a gene regulatory network. Leveraging privileged-feature distillation, it transfers knowledge from a teacher model that has access to a prior network. Overall, the approach is both technically sound and conceptually interesting.

**Questions:**

See above

**Ethical Concerns:**

["NO or VERY MINOR ethics concerns only"]

**Final Justification:**

The explanation by authors that incorporating prior may hurt the model's capacity to discover novel regulatory interactions is compelling and partly solve my problems.

Now I understand and agree with the rationale, but further stronger quantitative evidence may be even better to show incorporating prior may hurt the performance.

Overall, this rebuttal has addressed my main concerns. I will revise my score to borderline accept.

**Limitations:**

yes

**Quality:**

3

**Strengths And Weaknesses:**

**Strengths**

The paper proposes an interesting idea of using privileged feature distillation, which is used a lot in the recommendation system, both incorporating prior network knowledge and not limited by them. According to the results, the framework shows good performances.

**Weakness**

1. Though leveraging the idea of privileged distillation information is interesting, why do we have to distill that? In terms of the recommendation system, privileged features are not seen during the inference, since we have to recommend items before knowing the click rate. However, I didn’t observe the necessity of using it in regulatory network inference. In my understanding, we can still observe the prior network during inference. Is there any offline/online setting? Or any other reasons?

2. Following the above, if we look at Equation (4), it consists of both prediction loss and distillation loss. If the prior network is available during the inference, could the distillation loss be simply replaced by a regularization term that directly minimizes the distance between student learned A and prior M? It would be helpful if the authors could clarify whether they have any justification—either intuitive, theoretical, or empirical—for preferring distillation over direct regularization. Is there any additional information beyond the prior network M that is transferred from the teacher model during distillation?

3. In line 189-190, the author mentions that they select top-k from the adjacent matrix, and k is exactly the same as the ground truth. Is there any risk of information leakage, since the users have to first know the number of correct connections? Do other baselines rely on this setting? If setting a threshold (e.g., 0.5), does that still work well?

4. In terms of the writing, Equation (4) should be more clear. Inside the regulatory distillation loss, both $\theta_S$ and $\theta_T$ use same input, but teacher model should include an additional term representing prior network. In line 97, “hypnosis” should be “hypothesis”?

---

> ### Author Rebuttal · Authors · 2025-07-27
>
> **We sincerely appreciate your thoughtful and valuable feedback. We will address the concerns and questions point by point in the following responses.**
>
> ---
>
> ### ***Question 1***
> Though leveraging the idea of privileged distillation information is interesting, why do we have to distill that? In terms of the recommendation system, privileged features are not seen during the inference, since we have to recommend items before knowing the click rate. However, I didn’t observe the necessity of using it in regulatory network inference. In my understanding, we can still observe the prior network during inference. Is there any offline/online setting? Or any other reasons?
>
> ### ***Our response***
> ### **1. Clarification on prior knowledge usage during inference**
> We emphasize that our inference phase does not incorporate prior knowledge, consistent with the reviewer's observation that "*privileged features are not seen during inference*" in recommendation systems. As detailed in **Lines 54–56** of our manuscript, the student model is trained without prior knowledge. Further, **Lines 56–58** explicitly state that during inference, the student model **exclusively accepts gene expression data as input to infer a prior-free GRN**. Additionally, Figure 2 visually demonstrates that prior knowledge is solely provided to the teacher model, not the student model, and only gene expression matrices are input during inference.
>
> ### **2. Rationale for knowledge distillation**
> The motivation for distillation is elaborated in **Lines 30–43 and Figure 1**. Specifically, Inferring GRNs without priors involves an exponentially large search space (with $M$ genes, there are $M*(M−1)$ gene pairs to evaluate). While prior-based methods reduce this space to the $K$ edges provided in the prior knowledge to improve model performance, they introduce two critical issues:
> - The accuracy of the prior knowledge network will severely affect the model's performance, which depends on the overlap between the prior knowledge and the ground truth.
> - The inferred GRN is **strictly limited to a subset of the $K$ prior edges**, preventing discovery of real regulatory relationships outside this set. Given that priors are empirical and often contain high false-positive rates, this approach inherently caps model performance.
>
> **Therefore, we propose utilizing a distillation approach to address the limitations inherent in prior-based methods**. Our teacher model incorporates a prior knowledge network to learn comprehensive gene regulatory relationships. Subsequently, these learned features are input into the student model. **Crucially, the student model operates without requiring prior knowledge input. Instead, it learns the gene regulatory relationships by distilling knowledge from the features provided by the teacher model**, thereby enabling prior-free GRN inference. The self-attention computation within the student model operates without any masking mechanism (refer to the difference in the spatial layers between the teacher and student models in Figure 2). **This enables the student model to compute interactions between all possible gene pairs rather than be restricted in the $K$ edges from prior**. Since the student model does not receive prior knowledge input, it inherently avoids the two problems associated with prior networks. Furthermore, by employing the distillation strategy, we ensure the student model maintains robust inference accuracy.
>
> ---
>
> ### ***Question 2***
> Following the above, if we look at Equation (4), it consists of both prediction loss and distillation loss. If the prior network is available during the inference, could the distillation loss be simply replaced by a regularization term that directly minimizes the distance between student learned A and prior M? It would be helpful if the authors could clarify whether they have any justification—either intuitive, theoretical, or empirical—for preferring distillation over direct regularization. Is there any additional information beyond the prior network M that is transferred from the teacher model during distillation?
>
> ### ***Our response***
> ### **1. Clarification on prior knowledge usage during inference**
> As elaborated in our response to Question 1, the prior knowledge network is unavailable during the inference phase. The rationale for employing the distillation loss is also detailed therein.
>
> ### **2. Justification for preferring distillation over direct regularization**
> The regularization term suggested by the reviewers, which directly minimizes the distance between the adjacency matrix $A$ learned by the student model and the prior matrix $M$, represents a valid approach for model optimization. However, **we contend that the regularization term is unable to address the fundamental limitations inherent in prior-based methods (Lines 38-43)**. Specifically, the teacher model incorporates the prior knowledge network as an input during training, it utilizes this prior as a mask within the Transformer's self-attention mechanism. This mask constrains the attention calculation exclusively to gene pairs specified within the prior network. Consequently, if the student model's learned $A$ is forced to approximate the prior $M$, **the resulting regulatory relationships would remain confined to the $K$ edges defined by the prior**. This confinement does not resolve the core limitations of prior-based methods. Precisely for this reason, we adopted the distillation strategy. The student model does not receive the prior as input, instead, it learns from the features distilled by the teacher model. This approach enables the beneficial propagation of prior knowledge to the student while simultaneously circumventing the constraints imposed by the prior's limitations.
>
> ### **3. Temporal causality will be transferred to student model**
> Beyond the prior knowledge network $M$, the temporal causal relationships between genes learned by the teacher model are also distilled to the student model. The mechanism by which the teacher model learns these temporal causal associations through the temporal layer is detailed in **Lines 50-53 and 113-116**. Specifically, for time point $t$, a temporal mask is enforced to constrain each gene to attend only to information from preceding time steps ($< t$). Within the self-attention mechanism, attention weights between genes across these historical time steps are computed. Based on these weights, information from all time steps prior to $t$ undergoes weighted aggregation. This enables the representation of a gene at the current time step $t$ to integrate relevant historical context, thereby learning temporal dependencies among genes and ultimately leading to more accurate predictions of future gene expression.
>
> ---
>
> ### ***Question 3***
> In line 189-190, the author mentions that they select top-k from the adjacent matrix, and k is exactly the same as the ground truth. Is there any risk of information leakage, since the users have to first know the number of correct connections? Do other baselines rely on this setting? If setting a threshold (e.g., 0.5), does that still work well?
>
> ### ***Our response***
> We emphasize that selecting the top $K$ edges **constitutes the standard procedure for obtaining the final inferred GRN within the GRN inference field and all baseline methods rely on this setting**. Crucially, this procedure does not introduce information leakage. The selection occurs solely during the final inference stage, **the value of $K$ itself is not incorporated into the model's training process**. Its function is analogous to setting the number of clusters in clustering algorithms, which determines the final output structure without influencing the core learning algorithm.
>
> ---
>
> ### ***Question 4***
> In terms of the writing, Equation (4) should be more clear. Inside the regulatory distillation loss, both $\theta_{S}$ and $\theta_{T}$ use same input, but teacher model should include an additional term representing prior network. In line 97, “hypnosis” should be “hypothesis”?
>
> ### ***Our response***
> We sincerely thank the reviewer for identifying these two points. Specifically, we acknowledge that Equation (4) should explicitly reflect the teacher model's requirement for the prior knowledge network as an additional input, and we confirm that the term in Line 97 should correctly be "hypothesis". We apologize for these errors and confirm both will be rectified in the final manuscript.
>
> ---
>
> **We hope the responses provided above adequately address the reviewer's concerns. If there are any aspects that remain unclear, please do not hesitate to contact us further, we remain fully available to provide additional details.**

---

> > ### Comment · Reviewer_UYaC · 2025-08-01
> >
> > Thanks to the authors for detailed and thoughtful rebuttals. I understand the overall idea of using teacher-student framework on GRN, and recognize the effort to clarify the questions and limitations.
> >
> > I’d like to follow up on what I ask as the core question: the authors repeatedly emphasize that the student model cannot use the prior network during inference. However, the rationale seems not so clear. It’s very clear in recommendation systems, such as CTR or dwell time is unavailable during online inference, making the “training-time only privilege features” constraint necessary. But in the GRN settings, prior networks (e.g., Hi-C, ChIP-seq) are available both at training and inference time in many applications (correct if I’m wrong)
> >
> > Therefore, I’d like to clarity:\
> > **Is the choice of avoiding using prior in the student model because of a hard constraint in the real-world applications, or incorporating the prior would hurt performances?**\
> > If it’s the latter, then this is a strong and important point and should be emphasized. If it’s the former, could the authors provide more justification and real scenarios?
> >
> > In short, I’d like to believe the distillation approach itself is a reasonable beneficial method; but not very sure about the motivation behind it, not allowing the student model to use the prior at inference. Clarifying this would help strength the conceptual foundations of the paper.

---

> ### Author Response · Authors · 2025-08-02
>
> We thank the reviewer for raising this point. **We wish to emphasize that restricting the student model from using a prior network because its introduction will hurt model performance**.
>
> As highlighted, utilizing a prior network confines the model's exploration to the $K$ edges provided by the prior. Here, "model performance" specifically refers to its capability to discover novel regulatory relationships, a core competency critically valued within the GRN field for assessing a model's practical utility. Prior knowledge networks represent regulatory relationships already extensively validated through repeated experiments and literature reports. **In real-world biological applications, however, scientists are fundamentally more interested in identifying novel regulatory relationships**. This enables the discovery of new key transcription factors or target genes influencing diseases or developmental processes, thereby revealing unprecedented biological mechanisms, explaining the origins of complex phenotypes, or providing novel targets and intervention strategies for disease diagnosis and therapy.
>
> For example, analyzing gene expression data from cancer cells might reveal, through GRN inference, a previously unknown transcription factor regulating a novel pro-oncogenic gene network. This discovery could subsequently be validated through knockdown experiments targeting the new transcription factor to assess its therapeutic potential, ultimately leading to the development of precision therapies targeting this pathway.
>
> Relying solely on inference within validated prior knowledge inherently constrains a model's ability to discover novel regulatory interactions. Our distillation strategy is designed precisely to enable the student model to learn from prior knowledge while simultaneously retaining and enhancing its capacity to uncover new regulatory relationships.
>
> ---
>
> We sincerely appreciate your insightful feedback, which has significantly strengthened our work. We hope the responses provided comprehensively address the reviewer's concerns. Should any aspects require further clarification, we remain available to provide additional details upon request. If we have addressed your concerns, we would be grateful for your reconsideration of our manuscript. Thanks!

---

> ### Author Response · Authors · 2025-08-05
>
> We hope the responses provided adequately address the reviewer's concerns. We wonder if there are any aspects that remain unclear, if so, we remain fully available to provide additional details to enhance the manuscript. Thanks for your time and valuable feedback!

---

### Note · Authors · 2025-08-12

We sincerely thank the reviewer for the valuable feedback and constructive comments. Your expert insights have greatly helped us refine this work. We have carefully considered each question you raised, providing point-by-point responses, and will revise the manuscript according to your suggestions, which has significantly enhanced the paper's technical contributions and clarity of presentation.

KINDLE addresses a fundamental challenge in gene regulatory network inference: while prior knowledge improves prediction accuracy, it severely constrains the model's ability to discover novel regulatory relationships. This limitation prevents researchers from identifying key new genes and mechanisms in cellular development, disease progression, thereby directly impacting the development of new therapeutic targets and innovative treatment strategies.

KINDLE employs knowledge distillation where the teacher model learns from prior knowledge and transfers this knowledge to the student model. Crucially, the student model gains regulatory insights without being constrained by prior knowledge boundaries, maintaining the ability to explore the entire genome-wide regulatory space. This enables KINDLE to leverage existing biological knowledge while discovering previously unknown regulatory relationships.

KINDLE achieved superior performance across all benchmark datasets and demonstrated remarkable value in real biological applications. It successfully identified key transcription factors during mouse embryonic stem cell differentiation and accurately predicted cellular fate changes following Gata1 and Spi1 knockouts in hematopoietic stem cells, consistent with established experimental results, demonstrating KINDLE's capability to discover and model gene regulatory relationships.

We believe KINDLE provides a revolutionary paradigm that balances leveraging prior knowledge with discovering novel relationships. This approach holds particular value for studying rare diseases or insufficiently characterized biological processes, opening new possibilities for biological discovery.

---

### Decision · Program_Chairs · 2025-09-17

**Decision:**

Accept (poster)

**Comment:**

**Summary**
This paper introduces KINDLE, a teacher–student knowledge distillation framework for gene regulatory network (GRN) inference. The teacher model integrates prior biological knowledge with temporal expression data, while the student model learns through distillation and performs inference without priors. This design aims to capture the benefits of prior knowledge while retaining flexibility to discover novel interactions. The method is evaluated on benchmark datasets and shows performance gains over baselines, with case studies suggesting biological relevance.

**Strengths**
- Novel use of knowledge distillation to transfer information from a prior-informed model to a prior-free student model.
- Empirical results are generally strong, with state-of-the-art performance across several datasets.
- Biological case studies provide evidence that the method can yield meaningful insights.
- Authors engaged constructively in rebuttal and discussion, clarifying several technical points.

**Weaknesses**
- Some experimental concerns remain only partially addressed, including the need for additional ablations and clearer interpretability demonstrations.
- Biological validation, while promising, could be expanded for stronger support.

**Key reasons for decision**
The reviewers broadly agreed that the paper is technically sound and makes a novel contribution, with convincing empirical results. Remaining concerns about missing experiments and interpretability temper the enthusiasm but do not undermine the main contribution. Given this balance, a poster-level acceptance is warranted.

**Discussion and changes**
The rebuttal and discussion resolved some key questions. Reviewer UYaC’s concerns about excluding priors at inference were addressed by the authors’ explanation that priors limit discovery, which the reviewer accepted. Reviewer foG6’s questions on interpretability and feature transfer received detailed responses, though more empirical evidence would have been welcome. Reviewer 7DWh’s concerns on bias and evaluation were answered, but the review itself was not highly technical. Reviewer 2Z9D did not engage beyond an initial very brief Strong Accept despite multiple prompts, limiting the weight of that rating. Overall, the discussion improved confidence in the approach, but the uneven depth of reviewer engagement tempers the strength of the consensus.

**Decision: Accept (poster)**
The paper presents a novel and relevant framework with strong empirical results, though interpretability limitations and uneven review depth reduce confidence in a higher recommendation.